# Microphysical and thermodynamic phase analyses of Arctic low-level clouds measured above the sea ice and the open ocean in spring and summer

Manuel Moser[1,2], Christiane Voigt[1,2], Tina Jurkat-Witschas[2], Valerian Hahn[1,2], Guillaume Mioche[3], Olivier Jourdan[3], Régis Dupuy[3], Christophe Gourbeyre[3], Alfons Schwarzenboeck[3], Johannes Lucke[1,9], Yvonne Boose[4], Mario Mech[5], Stephan Borrmann[1,6], André Ehrlich[7], Andreas Herber[8], Christof Lüpkes[8], and Manfred Wendisch[7]

[1]Institut für Physik der Atmosphäre, Johannes Gutenberg-Universität, Mainz, Germany
[2]Institut für Physik der Atmosphäre, Deutsches Zentrum für Luft- und Raumfahrt, Wessling, Germany
[3]Laboratoire de Météorologie Physique, Université Clermont Auvergne, Clermont-Ferrand, France
[4]BreezoMeter, Haifa, Israel
[5]Institut für Geophysik und Meteorologie, Universität zu Köln, Cologne, Germany
[6]Particle Chemistry Department, Max Planck Institute for Chemistry, Mainz, Germany
[7]Leipziger Institut für Meteorologie, Universität Leipzig, Leipzig, Germany
[8]Alfred–Wegener–Institut, Helmholtz–Zentrum für Polar– und Meeresforschung, Bremerhaven, Germany
[9]Faculty of Aerospace Engineering, Delft University of Technology, Delft, Netherlands

**Correspondence:** Manuel Moser (manuel.moser@dlr.de)

**Abstract.** Airborne in-situ cloud measurements were carried out over the northern Fram Strait between Greenland and Svalbard in spring 2019 and summer 2020. In total, 811 minutes of low-level cloud observations were performed during 20 research flights above the sea ice and the open Arctic ocean with the Polar 5 research aircraft of the Alfred Wegener Institute. Here, we combine the comprehensive in-situ cloud data to investigate the distributions of particle number concentration N, effective diameter $D_{eff}$ and cloud water content CWC (liquid and ice) of Arctic clouds below 500 m altitude, measured at latitudes between 76 and 83°N. We developed a method to quantitatively derive the occurrence probability of their thermodynamic phase from the combination of microphysical cloud probe and Polar Nephelometer data. Finally, we assess changes in cloud microphysics and cloud phase related to ambient meteorological conditions in spring and summer and address effects of the sea ice and open ocean surface conditions. We find median N from 0.2 to $51.7\,\mathrm{cm}^{-3}$ and about two orders of magnitude higher N for mainly liquid clouds in summer compared to ice and mixed-phase clouds measured in spring. A southerly flow from the sea ice in cold air outbreaks dominates cloud formation processes at temperatures mostly below -10 °C in spring, while northerly warm air intrusions favor the formation of liquid clouds at warmer temperatures in summer. Our results show slightly higher N in clouds over the sea ice compared to the open ocean, indicating enhanced cloud formation processes over the sea ice. The median CWC is higher in summer ($0.16\,\mathrm{g\,m}^{-3}$) than in spring ($0.06\,\mathrm{g\,m}^{-3}$) as this is dominated by the available atmospheric water content and the temperatures at cloud formation level. We find large differences in the particle sizes in spring and summer and an impact of the surface conditions, which modifies the heat and moisture fluxes in the boundary layer. By combining microphysical cloud data with thermodynamic phase information from the Polar Nephelometer, we find

mixed-phase clouds as the dominant thermodynamic cloud phase in spring with a frequency of occurrence of 61% over the sea ice and 66% over the ocean. Pure ice clouds exist almost exclusively over the open ocean in spring, and in summer the cloud particles are most likely in the liquid water state.

The comprehensive low-level cloud data set will help to better understand the role of clouds and their thermodynamic phase in the Arctic radiation budget and to assess the performance of global climate models in a region of the world with strongest anthropogenic climate change.

# 1 Introduction

The impact of global warming is particularly strong in the Arctic, where temperatures rise at an accelerated rate relative to the rest of the globe, a phenomenon known as Arctic Amplification (Serreze and Francis, 2006). Clouds may play a key role for processes underlying the intense mean temperature rise in high latitudes (Wendisch et al., 2022). In contrast to cloud-free conditions, where the radiation energy budget is dominated by the low albedo of the dark open ocean, the presence of clouds significantly increases the reflection of solar radiation towards space and the emission of thermal-infrared radiation towards the surface. These changes of the atmospheric radiative energy budget are highly sensitive to the microphysical properties of Arctic clouds (Curry et al., 1996). In particular the size, shape, and thermodynamic phase of the hydrometeors influence the atmospheric energy fluxes and are often poorly represented in observations and models (Naud et al., 2014; Bodas-Salcedo et al., 2016; McCoy et al., 2016; Tan and Storelvmo, 2019; Wendisch et al., 2019; Kretzschmar et al., 2020). Observations show that the surface temperature is higher when clouds containing liquid water droplets are present (Shupe et al., 2022). Hence clouds have a direct impact on the sea ice thickness, snow depth, surface albedo, solar radiative energy input, and other parameters. In turn, the surface conditions feedback on the cloud properties (Stapf et al., 2020). Clouds frequently occur in the Arctic throughout the whole year (Mioche et al., 2015), for example, an occurrence of around 80 % was measured at the research station Ny-Ålesund, predominantly at altitudes below 2 km (Nomokonova et al., 2019). These low-level clouds are often found in a mixed-phase state (Shupe et al., 2006), representing a three-phase colloidal system consisting of water vapor, ice particles and coexisting supercooled liquid water droplets. In spite of many years of mixed-phase cloud research (Korolev et al., 2017) our knowledge about mixed-phase cloud physical processes remains incomplete. Hence, their representation in numerical weather prediction and climate models remains challenging (Morrison et al., 2011; Bock et al., 2020). In the Arctic the micro- and macrophysical properties of clouds are strongly affected by seasonal changes in meteorological weather situations such as atmospheric rivers, warm air intrusions, cold air outbreaks or Arctic cyclones, as well as small scale temperature and humidity fluctuations (McFarquhar et al., 2007; Mioche et al., 2017; Ruiz-Donoso et al., 2020; Wendisch et al., 2022). Turbulent fluxes and moisture transport are affected by the presence of sea ice (Lüpkes et al., 2011; Vihma et al., 2014; Wendisch et al., 2019; Elvidge et al., 2021; Schmale et al., 2021; Michaelis and Lüpkes, 2022). The total Arctic sea ice reaches its maximum extent in early March and a minimum in September which leads to a change in the overall surface properties, e.g. the surface albedo. During spring time a strong surface temperature gradient develops between sea ice and open ocean, while in summer the gradient is strongly reduced (Wendisch et al., 2022) which affects the structure of the lower atmosphere and, in particular,

the atmospheric boundary layer (ABL) and clouds within. In addition, different types of aerosol particles are formed and transported in the Arctic ABL and influence the cloud formation (Moschos et al., 2022).

Several studies have addressed the microphysical properties of low-level Arctic clouds measured by airborne in-situ observations before. Such studies often investigated the clouds in case studies at distinct meteorological situations or surface properties. McFarquhar et al. (2007) used airborne in-situ data to study the thermodynamic phase of Arctic clouds during fall. Results revealed that during a 4-day measurement period, clouds were mostly in a mixed-phase state, with a liquid layer at the top. Case studies by Lawson and Zuidema (2009) and Klingebiel et al. (2015) analyzed cloud particles using in-situ measurements. Lawson and Zuidema (2009) detected large dendrites, rimed ice and aggregates in summertime clouds formed in Arctic frontal and convective systems, while Klingebiel et al. (2015) examined liquid droplets in Arctic stratocumulus clouds in spring and found bimodal droplet size distribution at the cloud top. In-situ measured vertical Arctic cloud profiles during an airborne campaign in spring were analyzed by Mioche et al. (2017) regarding microphysical cloud properties. They found that the prevalent meteorological conditions had an impact on the cloud microphysical properties. Dodson et al. (2021) evaluated the microphysical properties of in-situ measured Arctic low-level clouds in September and compared these measurements with models. The study suggests that the observed discrepancy may be due to the models' poor representation of thermodynamic parameters. In a case study Young et al. (2016) investigated the microphysical properties of clouds during a cold air outbreak in March near the sea ice edge. The study revealed the strong near surface temperature increase as the primary driver of microphysical evolution during the transition from the sea ice to the open ocean.

It is essential to study microphysical cloud processes, in particular the properties of hydrometeors and the dominant thermodynamic cloud phase to improve our knowledge on Arctic Amplification and the Arctic radiative energy budget. During the two campaigns "Aircraft campaign observing FLUXes of energy and momentum in the cloudy boundary layer over polar sea ice and ocean" (AFLUX) and "Atmospheric airborne observations in the Central Arctic" (MOSAiC-ACA) conducted within the framework of the "Arctic Amplification: Climate Relevant Atmospheric and Surface Processes, and Feedback Mechanisms (AC)[3]" project (Wendisch et al., 2017), a comprehensive data set with microphysical measurements in low-level clouds was collected in the vicinity of Svalbard over the ice-covered and the open ocean in spring and in summer. In this work, we present an overview of low-level Arctic microphysical cloud properties and compare cloud particle number concentration, size and phase in spring and summer and for sea ice covered and open ocean conditions. We distinguish between liquid water, mixed-phase, ice clouds, and swollen aerosol particles, and present the frequency of occurrence of a certain thermodynamic phase depending on meteorological and surface conditions.

The article is structured as follows. In Sect. 2, the aircraft field campaigns are described including the meteorological situation using trajectory analysis, as well as the airborne in-situ instrumentation and the methodology of data evaluation. In Sect. 3.1 we distinguish the in-situ cloud particle measurements by the ambient atmosphere and surface conditions (spring over sea ice, spring over the open ocean, summer over sea ice and summer over ocean) and derive mean and altitude resolved microphysical cloud properties. By introducing a hydrometeor classification depending on particle number concentration, size and angular scattering properties, we study the microphysical properties and thermodynamic phase of low-level Arctic clouds

in Sect. 3.2 and discuss their frequency of occurrence. In Section 4 we summarize the findings of this study and discuss the implications.

## 2  Methods

### 2.1  The airborne field campaigns AFLUX and MOSAiC-ACA

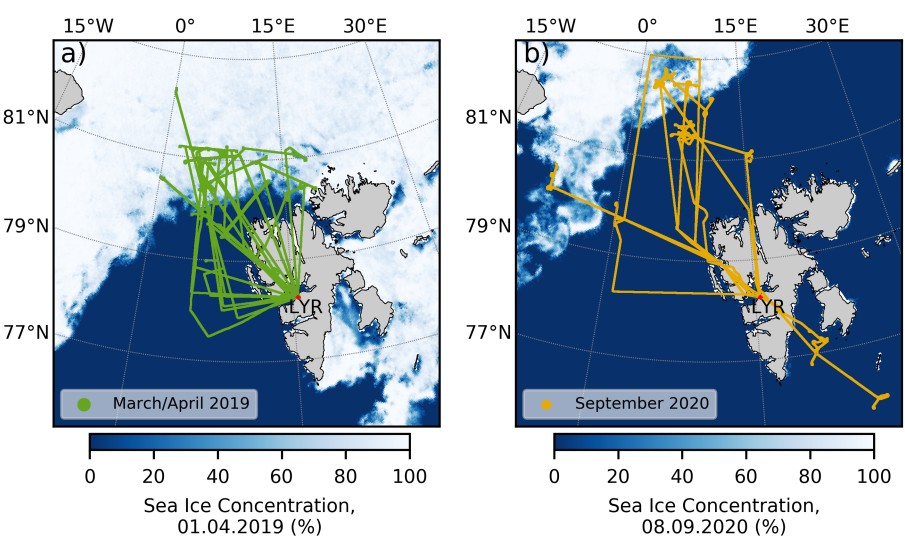

**Figure 1.** Maps of the flights during AFLUX (a), and MOSAiC-ACA (b) in the vicinity of Svalbard, Longyearbyen (LYR). Background shows the sea ice concentration at the halftime of each campaign recorded by the Advanced Microwave Scanning Radiometer 2 (AMSR2) onboard the GCOM-W1 satellite.

In-situ cloud data presented in this study were collected during the following two airborne field campaigns in the Arctic
region around Svalbard. The Aircraft campaign AFLUX was based in Longyearbyen (78°N, 015°E) and took place in the region of the Fram Strait in March and April 2019. The aircraft campaign MOSAiC-ACA, as part of the Multidisciplinary drifting Observatory for the Study of Arctic Climate (MOSAiC) expedition was conducted in September 2020 complementing the local atmospheric measurements on board of the German icebreaking research vessel RV Polarstern (Knust, 2017; Herber et al., 2021; Shupe et al., 2022).
During both campaigns, the research aircraft Polar 5, a former Douglas DC-3 specifically modified by Basler Turbo Conversions for flying under extreme polar conditions (BT-67; Wesche et al., 2016), operated by the Alfred Wegener Institute (AWI), was used as a platform to conduct remote sensing and in-situ measurements of clouds. A detailed description of the data collected during both campaigns is given by Mech et al. (2022a). The flight strategy for the campaigns was to provide both, in-situ and remote sensing measurements over the sea ice and the open ocean. The respective flight paths are displayed in
Fig. 1. The Fig. also shows the fraction of sea ice concentration (SIC) from GCOM-W1 satellite observations by the Advanced

**Table 1.** Flight Table for AFLUX summarizing the air mass origin (discussed in Sect. 2.2), the temperature range of cloud measurements, and the total time of cloud measurements. The fraction of measurements over the sea ice and open ocean are added. Note: Minutes in clouds over the sea ice and over the ocean do not add up to total, as for total all surface conditions are considered, for condition sea ice SIC > 80 % and condition ocean SIC < 20 % only.

| Date dd.mm.yyyy | Air mass origin | Cloud temperature range max / min (°C) | Minutes in cloud total | over sea ice / over ocean |
|---|---|---|---|---|
| 21.03.2019 | Ocean | -5.8 / -16.7 | 25.4 | 24.9 / 0.0 |
| 23.03.2019 | Sea ice | -13.1 / -23.3 | 68.4 | 44.3 / 0.0 |
| 24.03.2019 | Sea ice | -10.9 / -27.0 | 62.7 | 33.1 / 25.1 |
| 25.03.2019 | Sea ice | -10.2 / -28.3 | 61.3 | 50.1 / 7.2 |
| 30.03.2019 | Sea ice | -21.3 / -25.9 | 49.5 | 19.6 / 0.0 |
| 31.03.2019 | Sea ice | -13.8 / -26.4 | 54.2 | 11.7 / 33.6 |
| 01.04.2019 | Sea ice | -13.5 / -24.4 | 35.9 | 0.0 / 21.6 |
| 03.04.2019 | Sea ice | -13.7 / -21.8 | 32.0 | 24.3 / 3.5 |
| 04.04.2019 | Ocean | -5.5 / -14.7 | 14.7 | 0.1 / 13.6 |
| 06.04.2019 | Sea ice | -12.5 / -19.1 | 127.8 | 48.0 / 42.5 |
| 07.04.2019 | Sea ice | -13.9 / -17.8 | 48.7 | 1.1 / 5.5 |
| 08.04.2019 | Sea ice | -9.1 / -19.4 | 23.4 | 14.3 / 8.5 |
| 11.04.2019 | Sea ice | -1.8 / -19.0 | 52.8 | 52.5 / 0.1 |

**Table 2.** Flight Table for MOSAiC-ACA. Same columns as in Table 1.

| Date dd.mm.yyyy | Air mass origin | Cloud temperature range max / min (°C) | Minutes in cloud total | over sea ice / over ocean |
|---|---|---|---|---|
| 02.09.2020 | Sea ice | -1.0 / -1.7 | 1.5 | 0.0 / 1.5 |
| 04.09.2020 | Ocean | 13.6 / 4.7 | 41.1 | 0.0 / 41.1 |
| 07.09.2020 | Ocean | - | 0.0 | 0.0 / 0.0 |
| 08.09.2020 | Ocean | -1.4 / -4.0 | 20.9 | 4.1 / 11.2 |
| 10.09.2020 | Ocean | 2.6 / 0.1 | 36.0 | 12.8 / 21.2 |
| 11.09.2020 | Ocean | 0.2 / -3.1 | 18.4 | 3.9 / 7.6 |
| 13.09.2020 | Sea ice | -3.2 / -6.8 | 36.4 | 8.0 / 27.7 |

Microwave Scanning Radiometer 2 (AMSR2) instrument (Spreen et al., 2008), at a representative time for each campaign. The campaign periods were chosen to be in the season of largest and lowest sea ice extent in the Arctic. The dataset of this study consists of 1992 horizontal low-level in-situ cloud sequences (1685 during AFLUX and 307 during MOSAiC-ACA). One cloud sequence is defined as a continuous cloud measurement at the same altitude level. The cloud data covers more than
48668 measurement points at 1 Hz resolution (in total 811 min of cloud measurements, 657 min during AFLUX and 154 min

during MOSAiC-ACA) performed during 20 flights in the Arctic mainly within the ABL over sea ice and the open ocean (see Table 1 and 2). In the following Sect. 2.2 the meteorological conditions during both campaigns is explained using backward trajectory calculations.

## 2.2 Meteorological situation

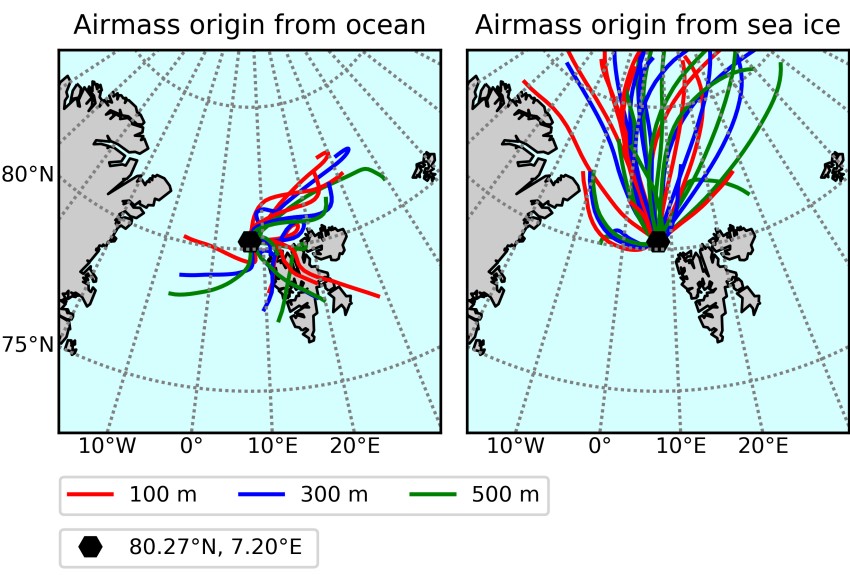

**Figure 2.** Air mass trajectories calculated based on 24-hour HYSPLIT backward analysis classified by dominant surface condition: Ocean (a) and sea ice (b).

The weather situations during both campaigns differed significantly. Colder temperatures in spring compared to summer and differences in Arctic sea ice extent have a major influence on the atmospheric temperature structure. Large-scale weather systems favor southerly air mass transport in the Fram Strait in spring, e.g. cold-air outbreaks, and northerly transport during summer, e.g. warm air intrusions and atmospheric rivers. Clouds form within a couple of hours inside the ABL. Also studies have shown that aerosol particle number concentration and chemical composition inside the Arctic ABL strongly depend on regional processes (Hartmann et al., 2020b; Köllner et al., 2021). In order to determine the origin of the probed air masses during both field campaigns, backward trajectories were calculated for each day with flights inside the ABL. Trajectories end at 100, 300 and 500 m altitude at a position representative for the low-level in-situ cloud measurements (80.27 °N, 007.20 °E). The air mass pathways were retrieved from the Hybrid Single-Particle Lagrangian Integrated Trajectory model (HYSPLIT) (Stein et al., 2015; Rolph et al., 2017) using the Global Data Assimilation System (GDAS) with 0.5 ° horizontal resolution as meteorological input for the AFLUX time period and the Global Forecast System (GFS) with 0.25 ° horizontal for the MOSAiC-ACA time period. In combination with the AMSR2 sea ice coverage, data for each flight day were classified with an air mass origin from the ocean or from the sea ice depending on the dominant surface condition below the air mass pathways

over the last 24 hours (see Fig. 2 and Table 1 and 2). Out of all considered cloud measurements inside the ABL 77.6% are attributed to air masses originating from the sea ice and 22.4% to air masses originating from the open ocean. During AFLUX

the general wind direction was dominated by off-ice direction while on-ice flow prevailed during summer (see Fig. 3a). Air masses classified as originating from the sea ice, can be attributed to cold air outbreaks in most cases. Here typically strong winds transport air masses over longer distances from the central Arctic to the Fram Strait within 24 hours. In contrast air masses originating from the ocean are mostly of regional origin, with the lowest observed latitude over the last 24 hours being 77.0 °N. Also, cloud temperature is strongly linked to the air mass origin with colder temperatures during off-ice flow compared

to on-ice flow. The temperature range for the low-level cloud measurements is shown in Fig. 3b and c, divided according to air mass origin and season.

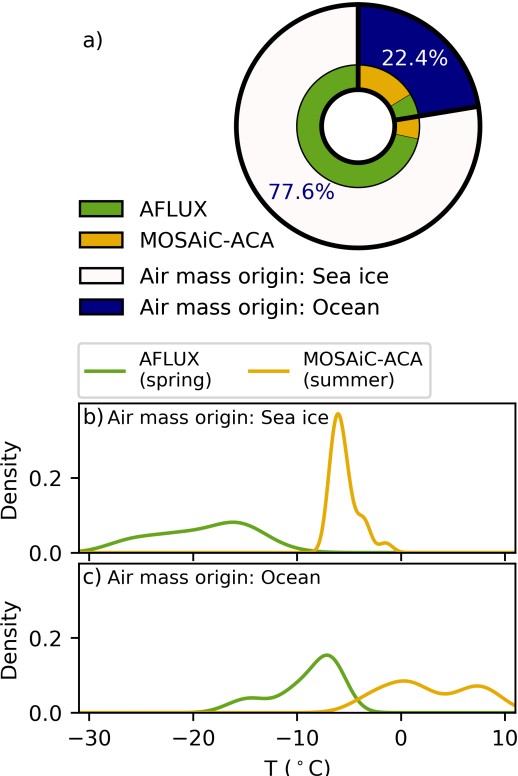

**Figure 3.** (a) Percentage of time with measurements inside clouds partitioned by air mass classified as originating from the ocean or sea ice and partitioned by the season spring (AFLUX) and summer (MOSAiC-ACA). Normalized frequency distribution of cloud temperatures in air masses originating from (b) the sea ice and (c) the open ocean during AFLUX and MOSAiC-ACA.

Although all flights were planned to avoid any influence of Svalbard, some trajectories indicate air masses overflowing the landmasses of the Svalbard archipelago. For these individual days the backwards trajectories in Fig. 2 might not be representative. The microphysical cloud properties presented in this section were measured along the Polar 5 flight track and depend on

the meteorological situation on these days. Climatological studies in the Fram Strait on air mass flow direction by Dahlke et al. (2022) are in line with our meteorological analyses for the seasons spring 2019 and summer 2020, supporting the hypothesis that the measured clouds can represent spring and summer conditions in the Arctic near Svalbard. Also the flight strategy remained the same for both campaigns, and with the large amount of cloud measurements over varying surface conditions and different flight days during both seasons reveal different microphysical properties for each season over the sea ice and over the

ocean.

## 2.3 In-situ instrumentation

During both campaigns, Polar 5 was equipped with an enhanced payload for in-situ cloud measurements, characterizing hydrometeors in a size range from 2.8 to 6400 μm. The instruments use two types of measurement techniques: scattering probes (Cloud Aerosol Spectrometer (CAS) for AFLUX, Cloud Droplet Probe (CDP) for MOSAiC-ACA and Polar Nephelometer

(PN)) and optical array probes (Cloud Imaging Probe (CIP), the Precipitation Imaging Probe (PIP) and the 2D Stereo Imaging Probe (2D-S)).

Data retrieved by the CAS and the CDP were used to derive the droplet size distribution from 2.8 to 50 μm, (e.g. Wendisch et al., 1996; Baumgardner et al., 2001, 2011; Wendisch and Brenguier, 2013; Kleine et al., 2018; Voigt et al., 2021). Both instruments count the number of cloud particles in the sampling volume and determine their individual size from the intensity

of forward scattered laser light (658 nm). Standard methods for calibration using mono-disperse glass beads were applied (Lance et al., 2010). The binning for the particle sizing was adopted using Mie theory with a refraction index of water (n = 1.33), including a distinct choice of bin limits to avoid ambiguities due to Mie resonances in the size range below 10 μm.

Particles in the size range of 30 μm to 6.4 mm are measured with optical array probes. The basic measurement principle of optical array probes consists of shadowgraphs of droplets and ice particles. Two-dimensional shadow images of hydrometeors

are reconstructed from individual image slices, where a slice monitors the state (shadowed or non shadowed) of a linear multi element photo diode array at a given moment in time. The data recorded by the CIP and the PIP (Baumgardner et al., 2001) differ in pixel resolution (CIP: 64 diode array with 15 μm resolution, PIP: 64 diode array with 103 μm resolution; Voigt et al., 2017). Data in the overlap size region are used to check the consistency between the cloud probes. The data processing includes identification and removal of shattered particle artifacts, stuck bit correction, and particle sizing which is done with

the processing software SODA (Software for OAP Data Analysis; Bansemer, 2023). The standard sizing method "circle-fit" is used for the particle diameter calculation which is defined as the diameter of the minimum enclosing circle of the projected 2D image. In addition to the CIP and PIP a 2D-S (Lawson et al., 2006), equipped with 128 diode array of 10 μm resolution was installed on the wing of Polar 5. The 2D-S data were used for a backup and validation of the CIP data.

The PN provides a direct measurement of the non-normalized scattering phase function (i.e. angular scattering coefficients,

ASC) of a volume of cloud particles crossing a collimated laser beam with a wavelength of 0.8 μm near the focal point of a parabolic mirror. The light scattered by water droplets, ice crystals or a mixture of both is recorded by a circular array of photodiodes (channels) (Gayet et al., 1997). Hence, the angular scattering pattern of cloud particle with diameter from a few micrometers to 1 mm can be obtained for scattering angles ranging from ±15 to ±162° and with an angular resolution of 3.5°.

Measurement errors lie between 3 to 5 % for scattering angles ranging from 15 to 155° with a maximum of 20 % at 162° (Shcherbakov et al., 2006). Averaged values of the calibrated ASCs were computed at a 1 Hz frequency and synchronized with the data recorded on the aircraft system. Electronic offsets of each channel were estimated and subtracted from the signal based on the signal measured during clear air sequences. Extinction coefficient and asymmetry parameter g can be derived from the ASC measurements (Gerber et al., 2000; Gayet et al., 2002, 2012) with uncertainties of ~25 % and ±0.04, respectively. Jourdan et al. (2003, 2010) showed that the combination of these parameters can be used to discriminate spherical from non-spherical cloud particles, as well as the dominant cloud thermodynamic phase.

All cloud probes were heated in order to avoid icing during the flights. Data are processed at 1 Hz frequency which corresponds to a spatial resolution between 50 and 90 m depending on the aircraft speed. The data recorded by the in-situ instrumentation during both campaigns are published in PANGAEA with open access (Moser and Voigt, 2022; Dupuy et al., 2022a; Moser et al., 2022; Dupuy et al., 2022b). Cloud data processing for both campaigns is explained in more detail by Mech et al. (2022a). The next section indicates how they are used to analyze the low-level clouds in both campaigns.

## 2.4 Processing of Arctic cloud data

Data presented in this study stem from a combination of three instruments: Particles below 30 µm diameter are exclusively detected by CAS or CDP. Between diameters of 30 to 40 µm, averaged data from the scattering probe and the CIP are calculated, from 40 to 250 µm diameter, CIP data only, in the overlap region 250 to 350 µm mean data of CIP and PIP and above 350 µm data recorded by the PIP are used. The microphysical cloud properties including the total particle number concentration (N), effective diameter ($D_{eff}$) and cloud water content (CWC) are calculated from the combined particle size distribution. In order to derive N, the number concentration for each particle size bin is added up. $D_{eff}$ is the ratio of the third to the second moment of the cloud particle size spectrum (Schumann et al., 2011). The CWC is defined here as the sum of the measured liquid and ice water content. Hydrometeors with diameters smaller than 50 µm are assumed to be droplets and those with diameters larger than 50 µm as ice which is appropriate for the majority of low-level Arctic mixed-phase clouds where ice dominates the large-particle regime (McFarquhar et al., 2007; Korolev et al., 2017). The ice water content is calculated using the mass-dimension relationship

$$m = a \times D^b \tag{1}$$

with D the particle diameter from the circle-fit method and the parameters (a = 0.00528 g cm$^{-b}$ and b = 2.1) proposed by Heymsfield et al. (2010, 2023). Another effective method to separate the liquid and ice fraction in clouds is recommended by D'Alessandro et al. (2019). The method classifies the thermodynamic phase of the cloud into ice, liquid or a mixed-phase based on a combination of microphysical properties recorded by similar in-situ cloud particle sizing instruments (Yang et al., 2021). In this work however the thermodynamic phase discrimination in Section 3.2 is achieved with the PN. Attributed to a recording failure of the CIP on 11 April 2019 the data was replaced by the 2D-S for that day. Uncertainties in cloud particle probe data depend on the microphysical cloud properties as certain particle size ranges are detected by different measurement techniques. In liquid and mixed-phase clouds N has a measurement uncertainty range of 10-30 % derived from the scattering

probes (Baumgardner et al., 2017). The larger ice crystals in ice clouds are counted by the optical array probes with an estimated uncertainty of approximately 50 % in N (Baumgardner et al., 2017; McFarquhar et al., 2017). In liquid clouds, the droplets are sized by the scattering probes, which have a range of 10-50 % uncertainty (Baumgardner et al., 2017). Sizing in ice and mixed-phase clouds is dominated by data from the optical array probes, which have an uncertainty of 20 % (Baumgardner et al., 2017; Gurganus and Lawson, 2018). CWC data have an uncertainty of 20 % for liquid clouds (Faber et al., 2018). In ice clouds, we assume an uncertainty of 50 % in CWC (Heymsfield et al., 2010; Hogan et al., 2012). In mixed-phase clouds we estimate the uncertainty of CWC to be in between liquid and ice cloud measurement, hence 20-50 %.

Basic meteorological parameters including wind, temperature, humidity and pressure along the flight track were provided by the meteorological instrumentation mounted at the nose boom of Polar 5. For position tracking GPS data is used. In this study, we restricted our measurements to horizontal in-situ flight legs to obtain microphysical cloud data with high statistical accuracy. The Arctic ABL over the sea ice is commonly quite thin and usually less than 500 m thick. Over the open ocean the ABL can extend to higher altitudes. However during both campaigns the majority of low-level in-situ cloud measurements were conducted below 500 m, to enable sufficient statistics for comparison. The cloud distribution versus altitude in Fig. 4, discussed in Sect. 3.1, supports our altitude threshold value.

In this study we distinguish the cloud data sets in four meteorological and surface conditions: Cloud data sets measured during AFLUX over the sea ice (spring-ice), during AFLUX over the ocean (spring-ocean), during MOSAiC-ACA over the sea ice (summer-ice) and during MOSAiC-ACA over the ocean (summer-ocean). For the surface characterization, we use AMSR2 satellite data with flight legs above sea ice (SIC > 80 %) and open ocean (SIC < 20 %). In order to avoid aerosol particles in Sect. 3.1 we define a cloud as a segment where the CWC exceeds $2 \times 10^{-4} \, \mathrm{g \, m^{-3}}$, while in Sect. 3.2 we do not set a threshold to account for all particles in the thermodynamic phase analyses.

# 3   Results and discussion

## 3.1   Microphysical properties of Arctic low-level clouds at different ambient meteorological and surface conditions

In this section, we analyze the measured microphysical cloud properties collected over sea ice and the open ocean during spring and summer and distinguish them in terms of seasons and surface conditions. A summary plot in Fig. 4 shows the variability of the CWC measured during both campaigns, AFLUX representing clouds in spring (c), and MOSAiC-ACA representing clouds in summer (d), as a function of altitude. Additionally the median temperature for the two seasons is shown in (e). Each circle in (c) and (d) represents a mean of a cloud measurement along one horizontal flight leg in clouds where successive 1 Hz datapoints fulfill CWC > $2 \times 10^{-4} \, \mathrm{g \, m^{-3}}$. This threshold removes more than 98 % of all measurements disturbed by aerosol particles, leaving, for example, the data of thick clouds, thin ice clouds, and the measurement of light precipitation untouched. The diameter of the datapoints indicates the mean N and the color transparency shows the duration of each continuous cloud measurement. Red datapoints correspond to clouds where surface conditions are classified as sea ice and blue corresponds to datapoints over open ocean. Box plots in (a) and (b) show the CWC data from (c) and (d) respectively, weighted by the measurement time within the cloud. The boxes represent the median, the upper and lower quartile and the whiskers give

the 97.5th and 2.5th percentile. Large differences between summer and spring clouds and clouds over sea ice and ocean are revealed. The median and percentiles of the measured low-level microphysical cloud properties in Fig. 4 over the sea ice

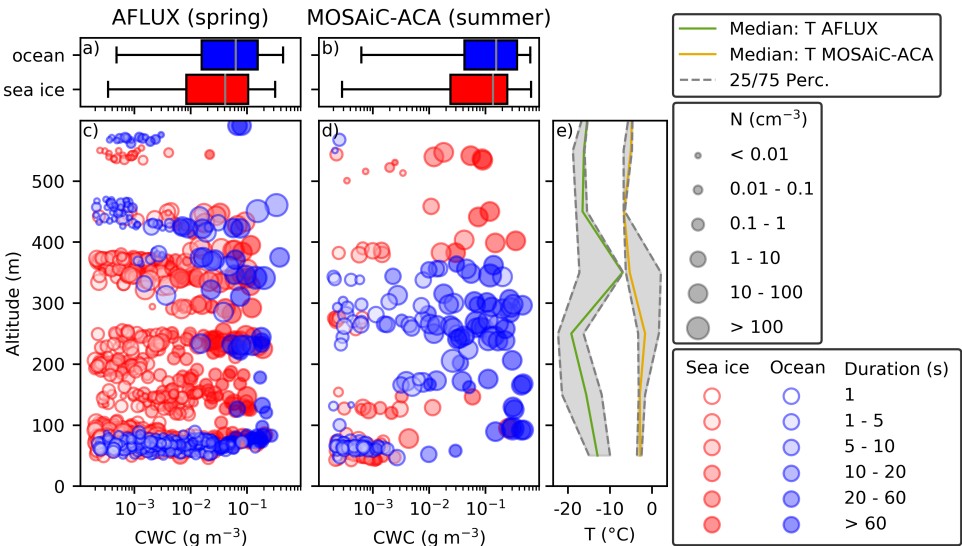

**Figure 4.** Overview of the in-situ measured low-level clouds during spring represented by AFLUX dataset (a and c) and summer represented by the MOSAiC-ACA dataset (b) and (d), depending on surface condition. (a) and (b) show all CWC measurements below 500 m in boxplots. (c) and (d) present the respective CWC values in altitude including information about N and duration of each cloud measurement. The median temperature for both seasons is shown in (e).

and the ocean during AFLUX and MOSAiC-ACA field campaign are given in Table 3. In addition to the microphysical cloud properties based on particles in the size range from $2.8\,\mu m$ to $6.4\,mm$, the microphysical cloud properties for liquid particles (based on particles $< 50\,\mu m$) and ice particles (based on particles $> 50\,\mu m$) only are presented. In order to determine whether

two values within a single column in Table 3 are statistically different, we conducted T-tests for the mean values and Wilcoxon tests for the medians. The significance level was set at 5% to decide whether the prevalent environmental conditions influence the properties of the clouds. We examined the following combinations for each property value within a row: Between the surface condition sea ice (i) and ocean (o) in spring (a) and in summer (m) (ia-oa, im-om), between spring and summer for the two surface conditions (ia-im, oa-om), as well as between the cloud data for each season (a-m) and surface condition (i-o). In

case there is a combination for which the difference is not statistically significant, it is marked with an asterisk in Table 3, and the corresponding combination is indicated in the caption. For example, the asterisk in the first row indicates that there is no significant difference in the data between the $\tilde{N}$ we observe for clouds over sea ice compared to cloud over the ocean during the summer campaign.

The largest differences of cloud properties are associated with the different seasons. Especially the medians of the $D_{eff}$

during summer are significantly reduced compared to spring, with values of $27\,\mu m$ over the sea ice and $33\,\mu m$ over the ocean in summer, compared to values of $403\,\mu m$ over sea ice and $1442\,\mu m$ over the ocean in spring. The main reason for this reduction

**Table 3.** Properties of Arctic low-level clouds ($< 500\,\mathrm{m}$) during AFLUX and MOSAiC-ACA for surface condition sea ice or ocean: Median number concentration $\tilde{N}$, median effective diameter $\tilde{D}_{\mathrm{eff}}$, median cloud water content $\tilde{\mathrm{CWC}}$ and mean horizontal cloud extent $\bar{d}_{\mathrm{cloud}}$ (calculated using the duration in cloud and mean aircraft speed, $V = 60\,\mathrm{m\,s^{-1}}$). The values in the square brackets give the 25th and 75th percentile respectively. The microphysical properties are calculated from all detected cloud particles as well as for particles smaller than $50\,\mu\mathrm{m}$ (assumed to be liquid) and for particles larger than $50\,\mu\mathrm{m}$ (assumed to be ice). An asterisk indicates that a combination of two values within this column is not significantly different. These combinations are as follows: $\tilde{N}$: im-om, $\bar{d}_{\mathrm{cloud}}$: ia-oa, ia-im, i-o, $\tilde{N}_{<50\mu\mathrm{m}}$: im-om, $\tilde{D}_{\mathrm{eff},<50\mu\mathrm{m}}$: ia-im, $\tilde{N}_{>50\mu\mathrm{m}}$: i-o.

|  | AFLUX (spring) | | MOSAiC-ACA (summer) | |
|---|---|---|---|---|
|  | sea ice | ocean | sea ice | ocean |
| $\tilde{N}$ (cm$^{-3}$)* | 0.70 [0.30 / 1.88] | 0.21 [0.07 / 0.57] | 51.72 [7.26 / 66.93] | 37.42 [13.94 / 65.80] |
| $\tilde{D}_{\mathrm{eff}}$ ($\mu$m) | 403 [161 / 924] | 1442 [807 / 2508] | 27 [19 / 32] | 33 [23 / 50] |
| $\tilde{\mathrm{CWC}}$ (g m$^{-3}$) | 0.04 [0.01 / 0.11] | 0.06 [0.02 / 0.16] | 0.14 [0.02 / 0.25] | 0.16 [0.04 / 0.37] |
| $\bar{d}_{\mathrm{cloud}}$ (m)*** | 1207 | 1313 | 1210 | 2670 |
| | | | | |
| $\tilde{N}_{<50\mu\mathrm{m}}$ (cm$^{-3}$)* | 0.65 [0.27 / 1.74] | 0.20 [0.06 / 0.56] | 51.68 [6.97 / 66.54] | 37.12 [13.66 / 65.22] |
| $\tilde{D}_{\mathrm{eff},<50\mu\mathrm{m}}$ ($\mu$m)* | 17 [6 / 34] | 10 [4 / 30] | 21 [15 / 25] | 22 [13 / 28] |
| $\tilde{\mathrm{CWC}}_{<50\mu\mathrm{m}}$ (g m$^{-3}$) | ( 1.74 [0.04 / 10.01] )$\times 10^{-3}$ | ( 0.14 [0.004 / 2.07] )$\times 10^{-3}$ | 0.12 [0.02 / 0.20] | 0.13 [0.02 / 0.26] |
| | | | | |
| $\tilde{N}_{>50\mu\mathrm{m}}$ (cm$^{-3}$)* | ( 6.0 [0.1 / 56.4] )$\times 10^{-3}$ | ( 2.5 [0.2 / 7.3] )$\times 10^{-3}$ | 0.14 [0.01 / 0.36] | 0.11 [0.002 / 0.60] |
| $\tilde{D}_{\mathrm{eff},>50\mu\mathrm{m}}$ ($\mu$m) | 627 [367 / 1340] | 1651 [979 / 2706] | 69 [66 / 83] | 72 [67 / 506] |
| $\tilde{\mathrm{CWC}}_{>50\mu\mathrm{m}}$ (g m$^{-3}$) | 0.02 [0.003 / 0.09] | 0.06 [0.01 / 0.15] | 0.02 [0.002 / 0.05] | 0.04 [0.01 / 0.09] |

is the ambient cloud temperatures in the respective seasons. As discussed in Sect. 2.2 cloud temperatures during summer campaign period are warmer compared to spring, with temperatures between -6.8 and +13.6°C during MOSAiC-ACA and between -28.3 and -1.8°C during AFLUX. In microphysical cloud analysis, it is important to consider the impact of seasonal
temperature variations. During the spring months, temperatures favor the growth of ice crystals, while temperatures above the freezing point during summer only allow for the existence of liquid cloud particles. As a result, the $D_{\mathrm{eff}}$ during spring correspond to ice crystals, while in summer, these values result from smaller liquid cloud particles. During summer at warmer temperatures, the median CWC is increased with a value of 0.16 g m$^{-3}$ over the ocean and 0.14 g m$^{-3}$ over the sea ice compared to colder conditions in spring, where the median CWC over the ocean is 0.06 g m$^{-3}$ and over the sea ice 0.04 g m$^{-3}$.
Also higher median N are found in summer, 51.7 cm$^{-3}$ over the sea ice and 37.4 cm$^{-3}$ over the ocean. In spring these values are reduced, 0.7 cm$^{-3}$ measured over the sea ice and 0.2 cm$^{-3}$ over the ocean. Similar to $D_{\mathrm{eff}}$, these changes in CWC and N can be traced back to the different temperature ranges and meteorological situations during both seasons. Greater CWC during MOSAiC-ACA is related to higher humidity and higher temperatures compared to cloud measurements during AFLUX. In spring, cold air outbreaks with strong winds from the central Arctic bring dry air with a low aerosol load. In contrast, in
summer, the weather situations favor the transport of moist air masses from the open ocean towards the sea ice. These different

synoptic situations impact cloud condensation nuclei and ice nuclei concentration, and thus influence N by cloud particle formation processes (Kirschler et al., 2022; Mech et al., 2020). The findings are supported by the microphysical cloud properties calculated for particles $< 50\,\mu m$ and $> 50\,\mu m$ in Table 3. Ice crystals dominate the $D_{eff}$ in spring and droplets dominate the $D_{eff}$ in summer. While droplets are the main contributor to the total CWC in summer, ice particles contribute most to the CWC in spring. The horizontal cloud extension is represented by the duration of a cloud measurement (mean aircraft speed at low-level cloud measurements at $60\,\mathrm{m\,s^{-1}}$) in Fig. 4. Mean low-level cloud length in summer over ocean is 2670 m (44.6 s) and over the sea ice less than half 1210 m (20.0 s). In spring horizontal cloud length are similar, 1313 m (21.8 s) over the ocean and 1207 m (20.3 s) over the sea ice. As a result of the ambient atmospheric conditions in spring, clouds are more patchy compared to summer. This may be due to strong winds, common in the Arctic spring, which may favor the formation of cumulus clouds in cold air outbreak weather situations. Warm air intrusions and frontal systems lead to larger cloud lengths in summer. The influence of different surface conditions on the horizontal cloud extension does not appear to be significant in our data. In Fig. 4 we observe a more homogeneous cloud distribution with altitude during the time of the spring campaign compared to summer. In spring, clouds are more equally distributed up to 450 m, with slightly smaller median CWC over sea ice compared to ocean. In summer, the CWC distribution with respect to altitude is more patchy and the measured cloud heights depend on the surface conditions. Over the ocean, most of the clouds were measured in altitudes between 220-380 m with CWC reaching $0.5\ \mathrm{g\,m^{-3}}$. The cloud layer in summer above the sea ice reaches to higher altitudes compared to the clouds above the ocean. However hardly any clouds are observed between 150-350 m. The more homogeneous vertical cloud distribution in spring and the more clustered and multilayered cloud structure in summer are in line with the visual observations made onboard the research aircraft. During AFLUX, one homogeneous cloud layer in the low-level altitude regime was regularly observed while most of the time during summer, a complex and patchy cloud structure with multiple individual layers was present. This can be associated to a stronger coupling of the ABL in spring time which leads to a well-mixed boundary layer. In summer, warm and moist air advection from the south leads to a stronger temperature inversion and favors multilayer clouds (Eirund et al., 2020). Besides the seasons, the prevalent surface conditions below the clouds show an impact on the microphysical cloud properties. A slightly greater CWC over the ocean is related to higher humidity and higher temperatures compared to cloud measurements over the sea ice. Over the warmer ocean compared to the sea ice, increased heat fluxes are induced and lead to a warmer and more turbulent ABL. Hence, an increased adiabatic liquid water content and the enhanced moisture transport into the ABL cause an increase of the cloud water content, allow hydrometeors to grow faster and lead to a deepening of the cloud layer. This also has an impact on particle growth rate. Due to the warmer and more turbulent ABL over the ocean, larger $D_{eff}$ are a result of a higher efficient collision-coalescence and subsequent growth via sustained supersaturation as explained by Young et al. (2016). This process could also explain the reduction of N over the ocean, which is significantly observed in spring. However, in Sect. 3.2 we will show, that the differences of N measured over the sea ice and the open ocean might result from different aerosol sources.

Consistent with Fig. 4 and Table 3 we find similar trends in altitude resolved cloud properties in Fig. 5. The altitude resolved profiles show lower N, larger $D_{eff}$ and lower CWC in spring compared to summer clouds. In Arctic spring clouds N increases with altitude to a maximum near 300 m and decreases near the top of the boundary layer. The large amount of particles, but

relatively small $D_{eff}$ at 300 m can be related to supercooled water droplets. Below, N decreases and $D_{eff}$ increases, which is due to ice crystals growing at the expense of supercooled water droplet and then precipitate. Lower CWC values at the upper cloud part can be explained by turbulent mixing and entrainment of dry air. Also during summer when clouds are most likely in a liquid state a decrease of $D_{eff}$ with altitude is observed. The larger median values of $D_{eff}$ at 300 m in summer can be addressed to the presence of ice crystals. Multilayer cloud structure and lower statistics in summer hammers the interpretation of altitude resolved summer clouds especially for N and CWC. Nevertheless, altitude resolved microphysical cloud parameters derived in spring are in line with the study of Lawson and Zuidema (2009) and Mioche et al. (2017) where the in-situ data were collected in vertical profiles.

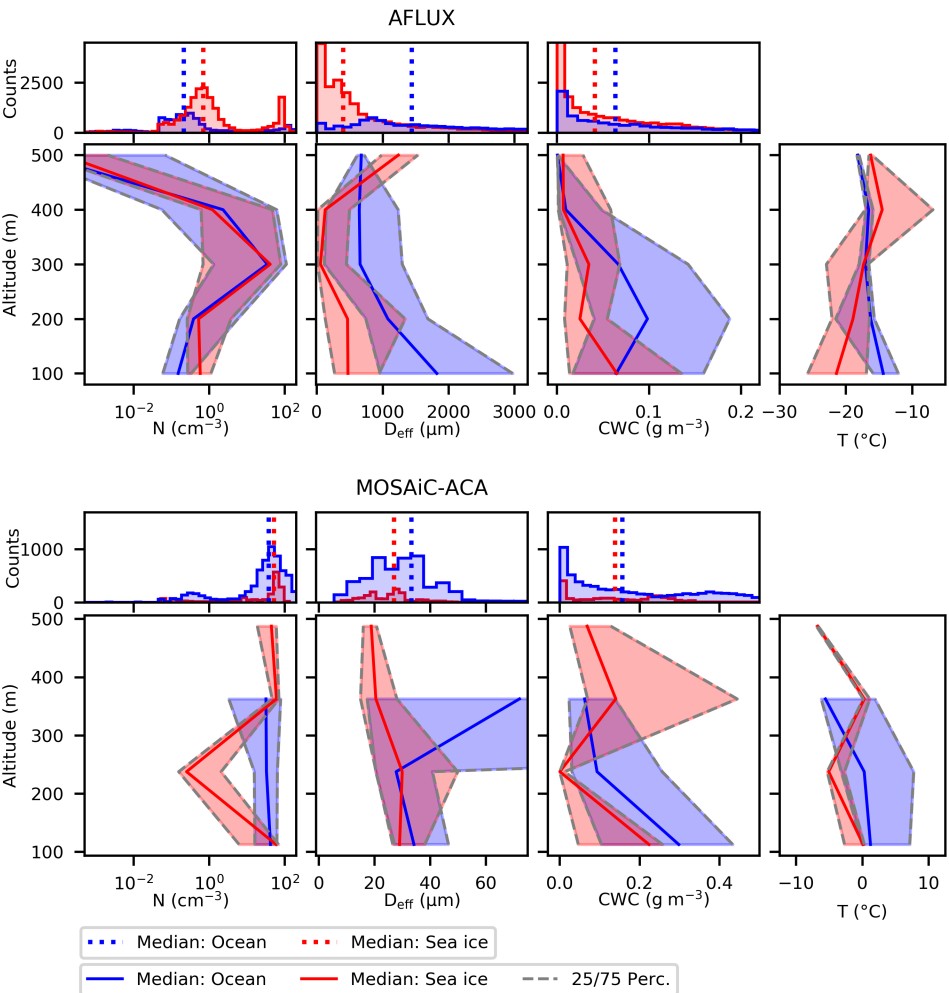

**Figure 5.** Height resolved microphysical cloud properties including N, $D_{eff}$ and CWC for condition spring (AFLUX), summer (MOSAiC-ACA), over sea ice and over the ocean.

## 3.2 Thermodynamic phase analyses of Arctic low-level clouds with respect to different ambient meteorological and surface conditions

In the following we discuss microphysical changes, including cloud particle size, concentration and thermodynamic phase, depending on surface structure and seasonal meteorological variations. All 1 Hz particle bulk measurements over the sea ice and the open ocean below 500 m for both campaigns are displayed in Fig. 6 in $D_{eff}$ versus N space. The color displays the number of 1 Hz particle measurements at the indicated $D_{eff}$ and N values. Peaks, which are areas with a higher probability of occurrence, are enclosed by rectangles, and in total 7 regimes are identified. The boundary values for each regime are bounded to include 80% of all data in one peak ($D_{eff}$ and N values between 10th to 90th percentile). Each peak of higher occurrence is separated by the minimum value to the neighboring peak. According to the prevailing thermodynamic phase, particle measurements in these regimes are classified as ice (1: 1a, 1b), mixed-phase (2: 2a, 2b, 2c), liquid (3) and aerosol particles (4). This classification is supported by particle size distributions from the combined particle measurement systems and by particle images by the CIP (Fig. 7) as well as by the asymmetry parameter and extinction coefficient measured with the PN (Fig. 8). In addition to the particle size distributions in Fig. 7, gamma functions are fitted over the sensitive size range of

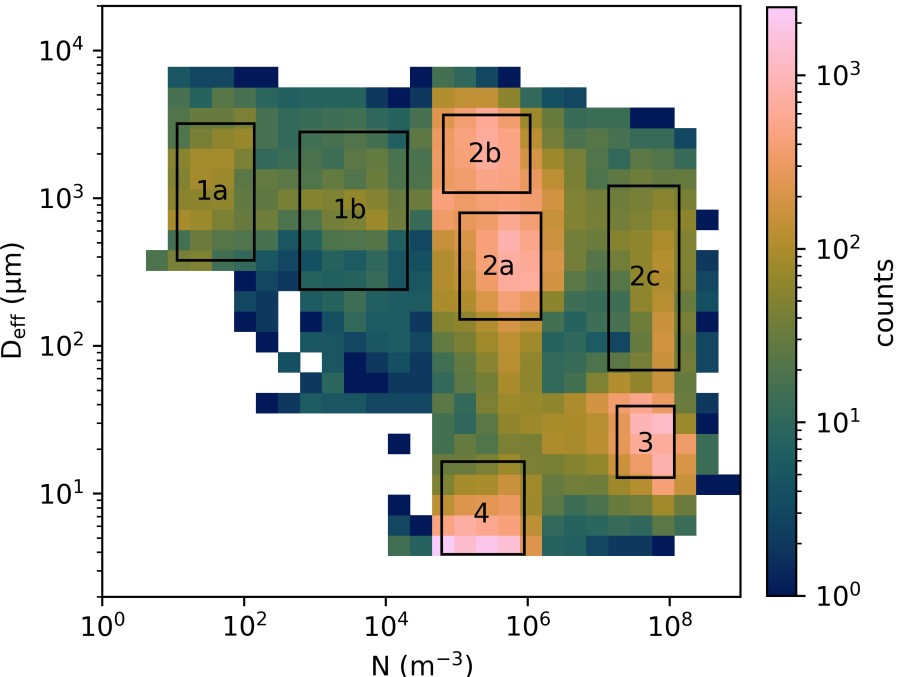

**Figure 6.** N versus $D_{eff}$ for 1Hz low-level particle data over sea ice and the open ocean ($< 500$ m) from the AFLUX and MOSAiC-ACA campaign combined, color coded by their occurrence. Regimes with increased occurrence frequency are marked with a rectangle. Associated cloud particles: 1 - ice, 2 - mixed-phase, 3 - liquid, 4 - aerosol particles

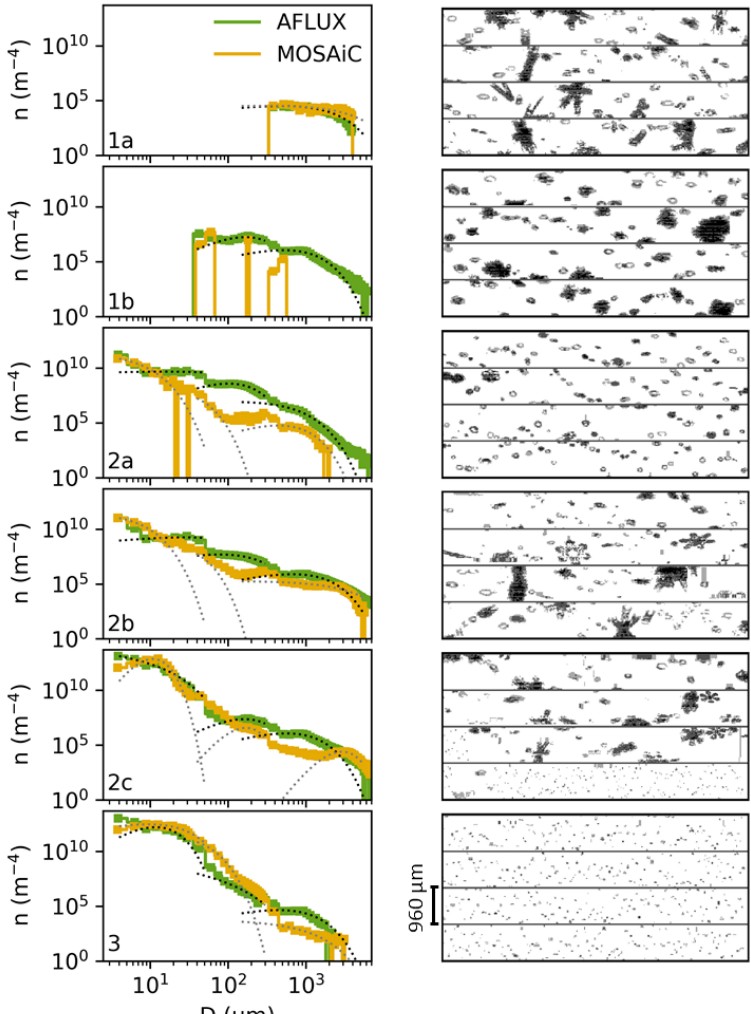

**Figure 7.** Particle size distribution and associated representative 2D images from CIP, for each rectangle cloud regime given by the Fig. 6.

the respective instrument. Cloud particle size distribution usually follow gamma type functions of the form:

$$N(D) = N_0 D^\mu e^{-\lambda D} \tag{2}$$

The fitted values for the dispersion μ, the slope $\lambda$ and the intercept $N_0$ are given in Table B1. Ice particles in the regimes 1a and 1b have low N and larger sizes. Regime 1a shows N between 11-140 $\mathrm{m}^{-3}$ with $D_{\mathrm{eff}}$ between 0.4-3.2 mm, and regime 1b shows higher N between 620-2×10$^4$ $\mathrm{m}^{-3}$ with diameters between 0.24-2.8 mm. Images from the CIP indicate that pristine ice crystals dominate for 1a, whereas graupel particles prevail in 1b (see Fig. 7). Region 1 measurements are precipitating ice particles from cloud layers above while a high fraction of 1b particles could have been in contact with a cloud layer where supercooled droplets were present facilitating the formation of graupel. Phase determination with the PN data remain

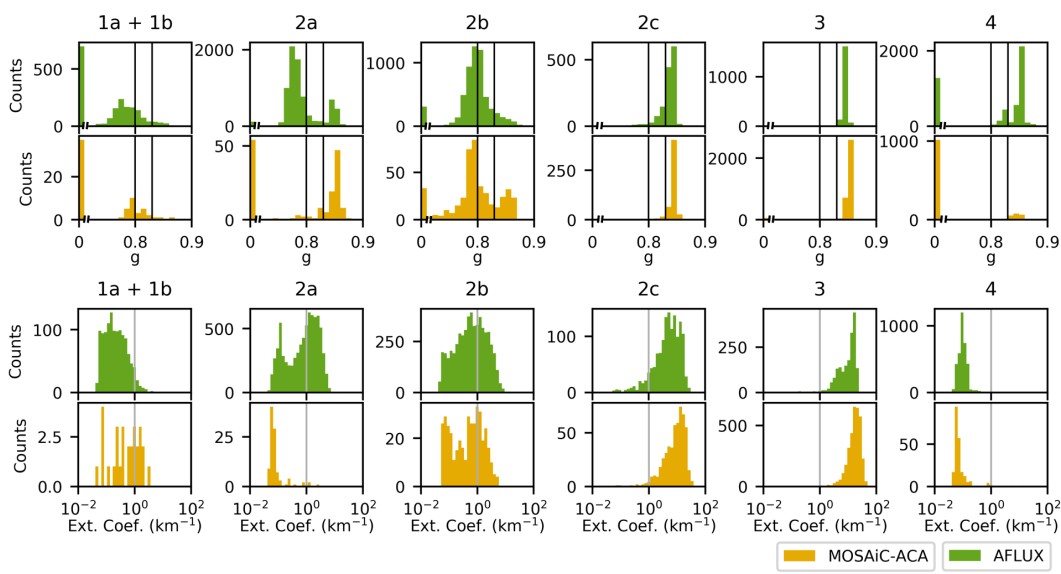

**Figure 8.** Frequency distribution of the asymmetry parameter and the extinction coefficient measured by the PN in each particle regime. Data is separated into measurements during AFLUX and MOSAiC-ACA.

challenging due to the very low N and large diameters of these ice crystals, often the PN does not detect any particles (extinction coefficient < 0.05 km$^{-1}$). Nevertheless, the remaining data show a clear trend towards ice phase, with g < 0.8 and extinction coefficient values between 0.1-1 km$^{-1}$.

Regime 2 indicated the presence of mixed phase clouds. In total three sub-divisions are addressed to regime 2: 2a (1.1×10$^5$ m$^{-3}$ < N < 1.5×10$^6$ m$^{-3}$ and 0.15 mm < D$_{eff}$ < 0.80 mm), 2b (6.5×10$^4$ m$^{-3}$ < N < 1.1×10$^6$ m$^{-3}$ and 1.1 mm < D$_{eff}$ < 3.6 mm) and 2c (1.4×10$^7$ m$^{-3}$ < N < 1.4×10$^8$ m$^{-3}$ and 0.07 mm < D$_{eff}$ < 1.2 mm). The N in these regimes are dominated by particles smaller than 40 µm and the D$_{eff}$ by the larger ice crystals. Data from the PN measurements reveal a mixed-phase state as g values cannot be clearly assigned to either liquid (g > 0.83) or ice phase (g < 0.8). Extinction coefficient ranges between 0.05 and 33 km$^{-1}$, intermediate values are typically observed for ice and liquid water. The individual regimes 2a and 2b differ slightly by the size of the ice crystals and the N of liquid droplets. Later we show 2a mixed-phase particles are frequently measured during AFLUX over the sea ice and 2b during AFLUX over the ocean. Thus, Arctic mixed-phase clouds over the ocean tend to have a slightly smaller N of liquid droplets and larger sizes of ice crystals compared to clouds over the sea ice in the same season. The air temperature, vertical wind velocities, humidity and aerosol particle concentrations impact the microphysical processes in the Arctic low-level mixed-phase clouds. The mixed phase clouds were measured at a mean temperature at -17.9°C which is close to -15 °C, where the maximum difference between water vapor partial pressure over ice and water is located. Such temperatures favor an enhanced ice crystal growth rate in mixed-phase clouds. The larger temperature gradient between open ocean and the atmosphere enhances vertical velocities and humidity transport which might induce a faster ice crystal growth rate. Similar to the slight increase of the total N in clouds (see Sect. 3.1) over the sea ice compared to the open ocean, the higher

number of liquid droplets in mixed-phase clouds over the sea ice could be explained by an increased cloud condensation nuclei concentration. Smaller ice crystals with higher N can be related to enhanced ice nucleating particle concentrations, which has

been observed in other studies in the central Arctic before (Hartmann et al., 2020a; Porter et al., 2022). Regime 2c, mixed-phase cloud measurements, is dominated by supercooled liquid cloud droplets with coexisting ice crystals, which have a higher N than 2a and 2b, as well as larger diameters compared to 2b. In all number 2 regimes the Wegener-Bergeron-Findeisen process (WBF; Wegener, 1912; Bergeron, 1935; Findeisen, 1938) is very likely, with different ice crystal and water droplet growth and evaporation rates.

Cloud data with $1.8 \times 10^7 \, \text{m}^{-3} < \text{N} < 1.2 \times 10^8 \, \text{m}^{-3}$, and $13 \, \mu\text{m} < \text{D}_\text{eff} < 40 \, \mu\text{m}$ are addressed to regime 3, cloud particles in liquid state. 2D images from the CIP show spherical particles while the extinction coefficient and asymmetry parameter from the PN support the assumption of the liquid phase (mean values: g = 0.84 and extinction coefficient = $16 \, \text{km}^{-1}$). Very rarely larger ice crystals may be present at the same time with liquid droplets. These particles are visible in the PSD, however negligible in the total particle concentration with > 5 orders of magnitude lower concentration than the liquid droplets.

Particle measurements with very small $\text{D}_\text{eff}$ (< 16 $\mu$m) and $6.2 \times 10^4 \, \text{m}^{-3} < \text{N} < 9.0 \times 10^5 \, \text{m}^{-3}$ are attributed to regime 4. These particles are too small to be resolved by the CIP and are exclusively recorded by the CAS and CDP. The PN data recorded here in this regime can not be addressed to any distinct cloud phase. Videos from onboard cameras show no visible clouds during regime 4 particle observations. We address these particles to large aerosol particles (lower detection limit of the particle measurement system: 2.8 $\mu$m) originated by the ocean and the sea ice.

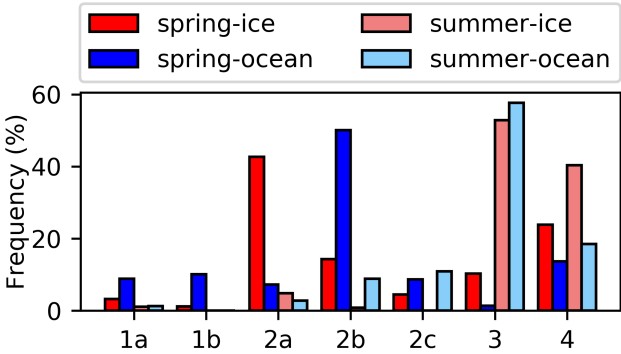

**Figure 9.** Frequency of occurrence for each particle regime (1a, 1b: Ice particles; 2a, 2b, 2c: Mixed-phase particles; 3: liquid particles; 4: Aerosol particles), separated by season and surface conditions. The values are normalized by the respective environmental conditions.

The frequency of occurrence of each particle regime with respect to the four conditions (spring-ice, spring-ocean, summer-ice and summer-ocean) is given in Fig. 9. In spring Arctic low-level clouds are most likely in a mixed-phase state (regimes 2a, 2b or 2c). The microphysics of mixed-phase clouds is slightly different depending on surface conditions as 2a mixed-phase clouds are measured with a higher frequency over the sea ice and 2b mixed-phase clouds dominate over the open ocean. Mixed-phase clouds with microphysics corresponding to 2c are observed with very low probability and are not found in summer time

over the sea ice. In general the mixed-phase state in clouds is suppressed in summer time as temperatures are too warm to

favor ice formation and the WBF process. Pure liquid phase (regime 3) is the prevailing cloud type in summer time, regardless of the surface. Temperatures close to 0 °C during the MOSAiC-ACA campaign (see Table 2) are warmer compared to spring and do suppress ice crystal formation or lead to melting of precipitating particles from above. So pristine ice clouds (1a and 1b) are mainly detected in spring, here with a higher probability over the ocean. Aerosols without liquid or ice phase particles

(regime 4) are frequently measured during both seasons with a higher frequency over the sea ice for both cases. An elevated N of small particles over the sea ice was already observed in previous studies, eg. in a case study by Young et al. (2016). Here the enhanced N is explained by swollen aerosol particles associated with a haze layer over the sea ice. As we observe this increase systematically in summer and spring where air mass origin differs strongly, we assume a local source. Such sources could be driven by biological processes in the sea ice (Dall´Osto et al., 2017; Hartmann et al., 2020a), however the presence of cracks,

open leads and polynyas in the sea ice have to be assumed. A more likely assumption are aerosols consisting of sea salt as the aerosol particles exceed diameters larger than 2.8 µm (Kirpes et al., 2018) (minimum size to be detected by the CDP). Over the ocean sea spray aerosols are emitted into the atmosphere via wave breaking mechanism (Blanchard, 1989). Over snow and ice-covered areas sea salt aerosols might be brought into the atmosphere by mechanisms related to blowing snow or frost flowers (Seguin et al., 2014; Xu et al., 2016; Yang et al., 2008; Huang and Jaeglé, 2017). But the mechanism of these processes

is still under discussion (Willis et al., 2018). In this respect, the data of AFLUX and MOSAiC-ACA propose that sea salt emitting processes over the sea ice are more efficient than over the open ocean. Sea salt aerosols can act as cloud condensation nuclei. Therefore, the higher number of sea salt aerosol particles over the sea ice could explain the enhanced N observed in Section 3.1 and the distribution of occurrence of 2a and 2b over the sea ice and the open ocean. Please note, the difference of Ñ between the surface ocean and sea ice in Table 3 does not pass a statistical significance test for the MOSAiC-ACA campaign.

However calculating Ñ for the dominant phase in summer, reveals a higher number of liquid particles over the sea ice during summer (see Table A3).

## 4    Summary and conclusion

During the two aircraft field campaigns, AFLUX and MOSAiC-ACA, we collected a comprehensive data set of microphysical cloud properties above the sea ice and the open ocean, representing low-level clouds in spring and summer in the Arctic.

We show that the microphysical cloud properties change significantly with seasonal meteorological and surface conditions. Primary results are listed below and schematically summarized in Fig. 10.

- In total we identify seven cloud regimes with different microphysical properties which we assign to four classes: Ice clouds, liquid clouds, a mixture of ice and liquid particles, and aerosol particles.

- Low-level ice clouds are exclusively observed over the ocean during spring. Due to warmer temperatures in summer,

clouds are most frequently in a liquid state.

- Mixed-phase clouds are the most prevalent state for clouds in spring.

- Median N is enhanced by two orders of magnitude during summer compared to spring caused by the different meteorological situations which favor liquid phase clouds in summer.

- The median CWC is increased by more than a factor of 2 during summer compared to spring and appears enhanced in both seasons over the open ocean compared to measurements above the sea ice due to a warmer and more humid atmosphere.

- We observe larger ice crystals in spring and smaller liquid droplets in summer conditions.

- Slightly enhanced $D_{eff}$ and CWC over the ocean compared to cloud measurements above the sea ice result from a more turbulent ABL and increased heat fluxes.

- The increased N observed in mixed-phase clouds, in aerosol particles, in liquid clouds and in the total cloud particle measurements above the sea ice may be explained by surface processes emitting sea salt.

- The horizontal cloud length quantified by the horizontal extension of the low-level cloud encounters grows with rising temperature and humidity within the ABL and is largest in summer.

This work provides a direct comparison of microphysical cloud properties and cloud phases related to surface conditions during the seasons of maximum and minimum sea ice extent over the Arctic Ocean. The comprehensive observations can help to evaluate satellite retrievals of Arctic low-level clouds. Our cloud data can be used to develop and evaluate parameterizations of Arctic clouds in process models and to better understand the influence of different meteorological and surface conditions on clouds. Improving the representation of microphysical cloud properties and their radiative impact in global climate models may elucidate the role of clouds for Arctic Amplification and for future climate change.

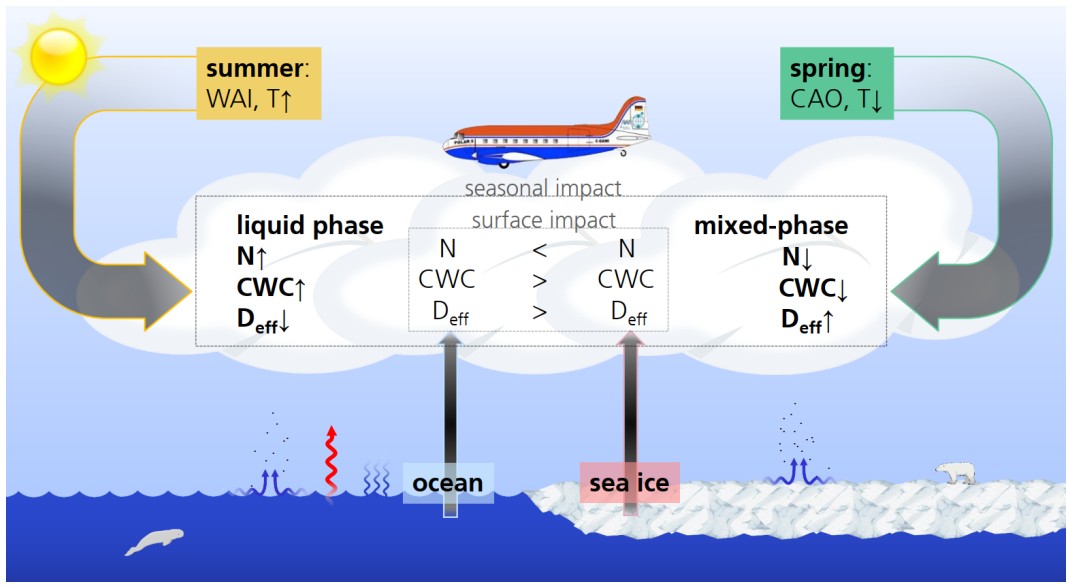

**Figure 10.** Schematic representation of the results: Black arrows symbolize the influence on the microphysical properties of Arctic low-level clouds, which is most pronounced during different seasons. In both seasons, the prevailing surface condition modify the microphysical cloud properties due to regional atmospheric-surface processes. Abbreviations: WAI - "Warm Air Intrusion", CAO - "Cold Air Outbreak".

 **Appendix A: Microphysical properties separated by the thermodynamic phase**

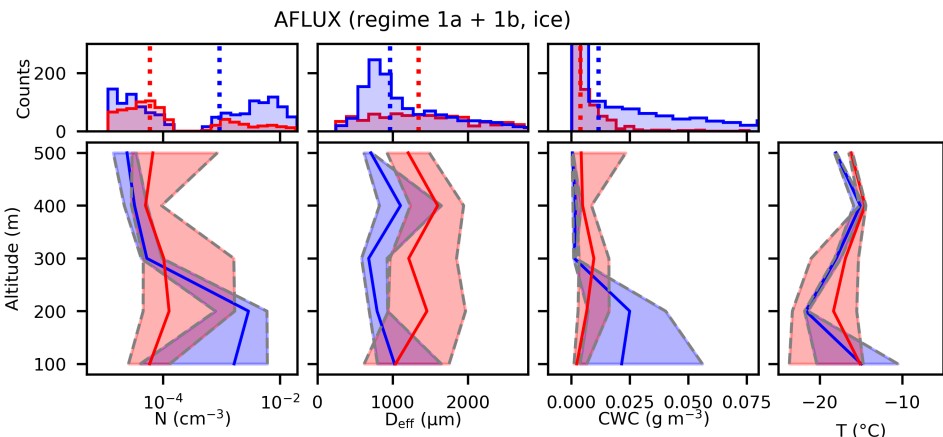

**Figure A1.** Same presentation of height resolved microphysical properties in Fig. 5, but for cloud data from regimes 1a and 1b, classified as ice phase. Hardly any ice phase was measured during MOSAiC-ACA. Corresponding median values are given in Table A1.

**Table A1.** Similar to Table 3 but with cloud data from regimes 1a and 1b, classified as ice phase.

|  | AFLUX (spring) | |
|---|---|---|
|  | sea ice | ocean |
| $\tilde{N}$ (cm$^{-3}$) | $(0.06\,[0.03\,/\,0.82])\times 10^{-3}$ | $(0.84\,[0.03\,/\,4.87])\times 10^{-3}$ |
| $\tilde{D}_{eff}$ (μm) | 1339 [829 / 1875] | 956 [732 / 1545] |
| $\tilde{CWC}$ (g m$^{-3}$) | $(3.9\,[1.7\,/\,8.8])\times 10^{-3}$ | $(10.8\,[1.6\,/\,44.4])\times 10^{-3}$ |

**Table A2.** Similar to Table 3 but with cloud data from regimes 2a, 2b and 2c, classified as mixed-phase. Differences within one column were tested for significance with method used in Sect. 3.1.

|  | AFLUX (spring) | | MOSAiC-ACA (summer) | |
|---|---|---|---|---|
|  | sea ice | ocean | sea ice | ocean |
| $\tilde{N}$ (cm$^{-3}$) | 0.56 [0.32 / 0.90] | 0.29 [0.17 / 0.53] | 0.15 [0.13 / 0.24] | 0.93 [0.36 / 62.03] |
| $\tilde{D}_{eff}$ (μm) | 455 [282 / 762] | 1730 [1120 / 2546] | 384 [249 / 576] | 584 [227 / 2459] |
| $\tilde{CWC}$ (g m$^{-3}$) | $(50.7\,[12.9\,/\,107.2])\times 10^{-3}$ | $(99.4\,[35.7\,/\,194.4])\times 10^{-3}$ | $(0.6\,[0.2\,/\,1.4])\times 10^{-3}$ | $(44.3\,[16.4\,/\,88.6])\times 10^{-3}$ |

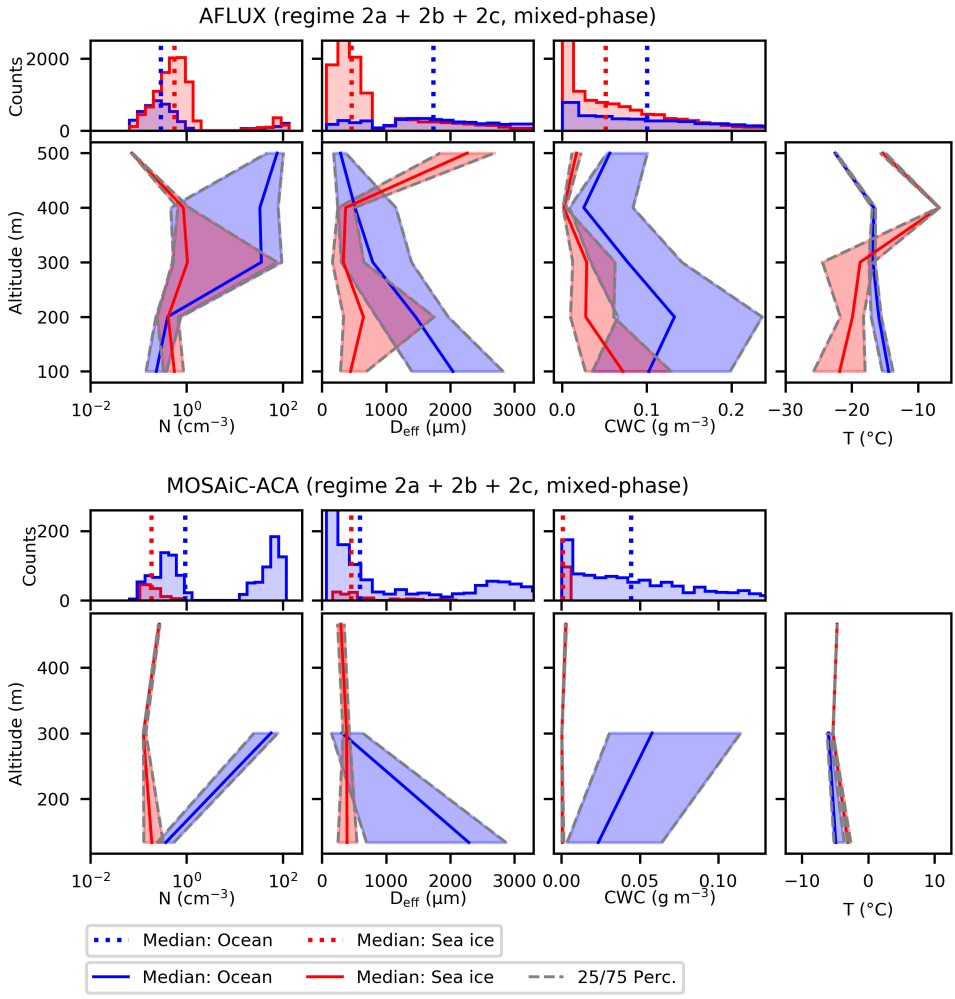

**Figure A2.** Same presentation of height resolved microphysical properties in Fig. 5, but for cloud data from regimes 2a, 2b and 2c, classified as mixed-phase. Corresponding median values are given in Table A1.

**Table A3.** Similar to Table 3 but with cloud data from regime 3, classified as liquid phase. Differences within one column were tested for significance with method used in Sect. 3.1.

| | AFLUX (spring) | | MOSAiC-ACA (summer) | |
|---|---|---|---|---|
| | sea ice | ocean | sea ice | ocean |
| $\tilde{N}$ (cm$^{-3}$) | 76.77 [52.11 / 85.53] | 83.27 [52.41 / 99.85] | 64.13 [54.82 / 71.84] | 47.68 [34.66 / 61.80] |
| $\tilde{D}_{eff}$ (µm) | 17 [15 / 24] | 24 [18 / 32] | 25 [19 / 28] | 30 [23 / 34] |
| $\tilde{CWC}$ (g m$^{-3}$) | 0.08 [0.04 / 0.12] | 0.03 [0.02 / 0.06] | 0.23 [0.13 / 0.28] | 0.31 [0.15 / 0.46] |

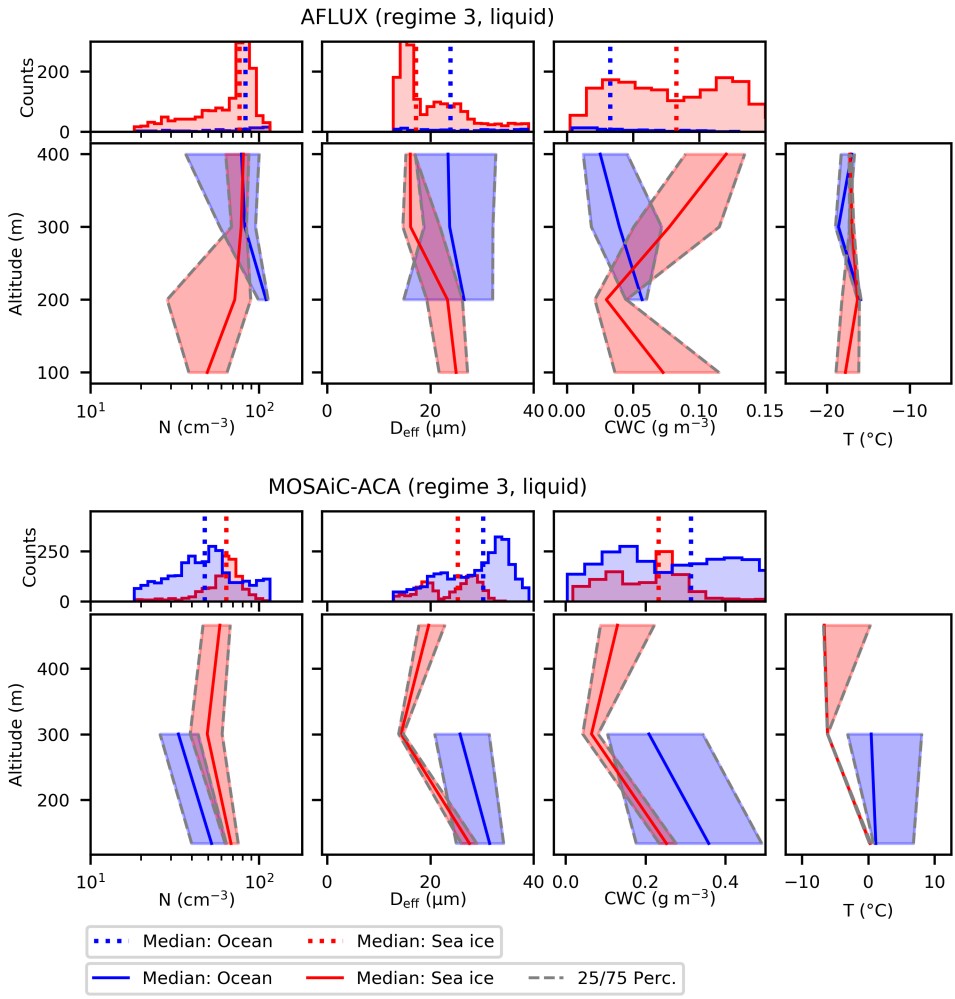

**Figure A3.** Same presentation of height resolved microphysical properties in Fig. 5, but for cloud data from regimes 3, classified as liquid phase. Corresponding median values are given in Table A3.

**Table A4.** Similar to Table 3 but with data from regime 4, classified as aerosol particles ($> 2.8\,\mu m$). The asterisk indicates that one combination of two values within the column is not significantly different: im-om.

| | AFLUX (spring) | | MOSAiC-ACA (summer) | |
|---|---|---|---|---|
| | sea ice | ocean | sea ice | ocean |
| $\tilde{N}\ (cm^{-3})*$ | 0.38 [0.21 / 0.56] | 0.21 [0.12 / 0.31] | 0.13 [0.12 / 0.22] | 0.13 [0.12 / 0.25] |

# Appendix B: Supplementary to Section 3.2

**Table B1.** Fitted parameters for the gamma functions in Fig. 7 for the size range of the CDP/CAS, CIP and PIP. The index CAS represents both instruments, CAS and CDP.

| regime | campain | $N_{0,\text{CAS}}$ $(m^{-4-\mu})$ | $\mu_{\text{CAS}}$ () | $\lambda_{\text{CAS}}$ $(m^{-1})$ | $N_{0,\text{CIP}}$ $(m^{-4-\mu})$ | $\mu_{\text{CIP}}$ () | $\lambda_{\text{CIP}}$ $(m^{-1})$ | $N_{0,\text{PIP}}$ $(m^{-4-\mu})$ | $\mu_{\text{PIP}}$ () | $\lambda_{\text{PIP}}$ $(m^{-1})$ |
|---|---|---|---|---|---|---|---|---|---|---|
| 1a | AFLUX | - | - | - | - | - | - | $2.42\times10^{7}$ | 0.77 | 1559.47 |
| 1a | MOSAiC | - | - | - | - | - | - | $2.03\times10^{5}$ | 0.20 | 706.94 |
| 1b | AFLUX | - | - | - | $1.48\times10^{23}$ | 3.79 | $2.24\times10^{4}$ | $9.50\times10^{12}$ | 1.86 | 3581.95 |
| 1b | MOSAiC | - | - | - | - | - | - | - | - | - |
| 2a | AFLUX | $3.94\times10^{10}$ | 0.18 | $1.11\times10^{4}$ | $2.32\times10^{22}$ | 3.11 | $3.06\times10^{4}$ | $5.15\times10^{08}$ | 0.41 | 3912.93 |
| 2a | MOSAiC | $1.61\times10^{13}$ | 0.30 | $4.21\times10^{5}$ | $4.28\times10^{19}$ | 2.27 | $1.39\times10^{5}$ | $2.74\times10^{14}$ | 2.56 | 5774.83 |
| 2b | AFLUX | $2.11\times10^{12}$ | 0.62 | $2.19\times10^{4}$ | $5.07\times10^{13}$ | 1.33 | $1.67\times10^{4}$ | $5.28\times10^{11}$ | 1.62 | 2395.79 |
| 2b | MOSAiC | $9.71\times10^{15}$ | 0.75 | $5.22\times10^{5}$ | $3.60\times10^{23}$ | 2.93 | $1.70\times10^{5}$ | $2.41\times10^{5}$ | 0.03 | 642.56 |
| 2c | AFLUX | $6.59\times10^{8}$ | -0.84 | $1.48\times10^{5}$ | $4.88\times10^{24}$ | 4.11 | $2.50\times10^{4}$ | $4.79\times10^{13}$ | 2.07 | 3671.13 |
| 2c | MOSAiC | $2.09\times10^{81}$ | 12.69 | $1.17\times10^{6}$ | $1.98\times10^{51}$ | 10.64 | $6.26\times10^{4}$ | $9.58\times10^{35}$ | 10.64 | 3635.66 |
| 3 | AFLUX | $2.85\times10^{39}$ | 5.08 | $4.31\times10^{5}$ | 541.68 | -1.26 | $1.56\times10^{4}$ | $4.90\times10^{10}$ | 1.60 | 3585.70 |
| 3 | MOSAiC | $1.97\times10^{20}$ | 1.43 | $1.54\times10^{5}$ | $2.65\times10^{20}$ | 1.76 | $1.06\times10^{5}$ | 2231.82 | -0.10 | 2206.11 |

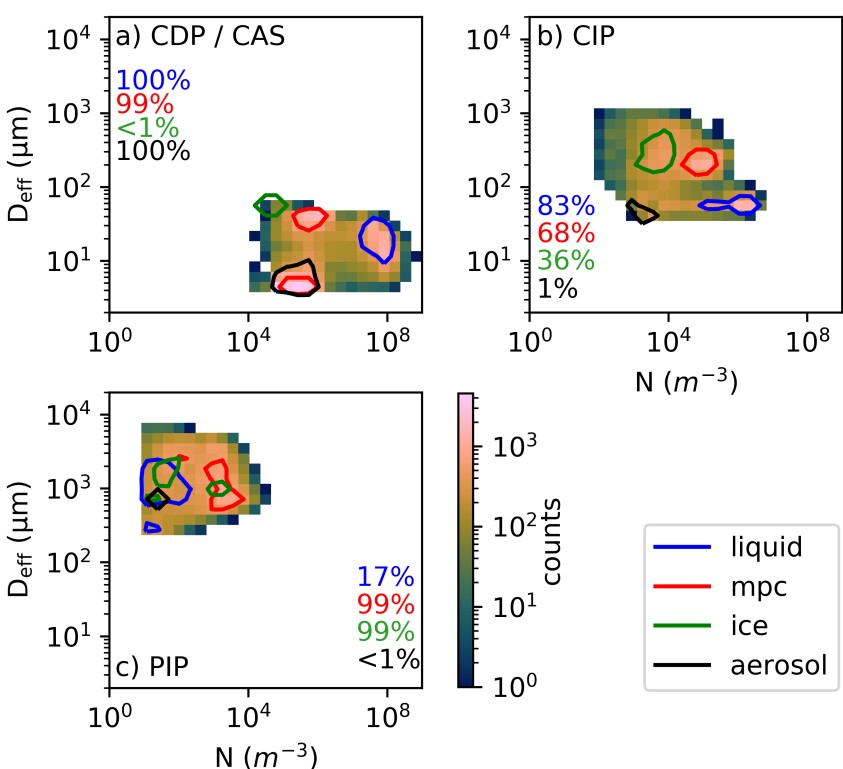

**Figure B1.** N versus $D_{eff}$ as presented in Fig. 6 measured by the individual particle probes. The contour line encloses 95% of the detected measurements classified as liquid, mpc ice, and aerosol by the regimes in Fig. 6. The percentage values indicate the proportion of measurements detected by the respective instruments.

*Data availability.* Processed in-situ data from the AFLUX and MOSAiC-ACA campains are freely available via the world data center PANGAEA (Moser and Voigt, 2022; Moser et al., 2022; Dupuy et al., 2022a, b). Data can be easily reproduced and analyzed by the python package *ac3airborne* (Mech et al., 2022b) including a package for flight segmentation (Risse et al., 2022), where each research flight is split
up into logical parts like ascends, descends, specific patterns for in-situ probing, etc.. The data ac3airborne package provides as well access to sea ice coverage along the flight path extracted from data available at University of Bremen ( https://seaice.uni-bremen.de/). Raw in-situ cloud data recorded by the CAS, CAS, CIP and PIP are archived at the German Aerospace Center and are available on request. Raw data by the PN and 2D-S are available from R. Dupuy (regis.dupuy@uca.fr) on request. The HYSPLIT model is a freely accessible online tool available at https://www.ready.noaa.gov/HYSPLIT.php. Figures have been designed with the python software Pylustrator (Gerum, 2020).

*Author contributions.* MMo conducted the analysis and wrote the manuscript. CV and TJW supervised the study and provided intensive feedback on the manuscript. MMo, VH, GM, OJ, RD, CG, AS, JL, YB and SB have been responsible for the in-situ probes and performed the measurements during both campaigns. MMe, AE, AH, CL, and MW conceived the flight experiments. MMo processed the data of the CAS, CDP, CIP and PIP. RD processed the data recorded by the PN and the 2D-S probe. OJ and GM analyzed the PN data in Sect. 3.2. All authors reviewed the manuscript and added valuable suggestions to the final draft.

*Acknowledgements.* We gratefully acknowledge the funding by the German Reseach Foundation (DFG, Deutsche Forschungsgemeinschaft) under the Priority Program SPP PROM Vo1504/5-1, the Transregional Collaborative Research Center SFB/TRR 172 (Project-ID 268020496) and the SPP1294 HALO under Vo1504/7-1.

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
