# Peer review of "Microphysical and thermodynamic phase analyses of Arctic low-level clouds measured above the sea ice and the open ocean in spring and summer"

_Atmospheric Chemistry and Physics, 2023_

## Referee Comment (RC1)

Review of "Microphysical and thermodynamic phase analyses of Arctic low-level clouds measured above the sea ice and the open ocean in spring and summer", by Moser and coauthors, ACP-2023-44.

This is a very comprehensive article that compares the cloud microphysical properties in spring and summer seasons in low-level clouds over the Arctic ice and open water regions. The field programs sampled with the Polar 5 instrumented aircraft are AFLUX and MOSAiC-ACA. The properties of the clouds sampled during the two field programs differ because of the temperatures involves, the amount of sea ice sampled during each period, and the CCN/IN contents. Overarching reasons for conducting this analysis is to gain a better understanding of the arctic radiation budget-seasonally, and to provide data for use in the evaluation of climate model representations of the arctic radiation budget. Overall, this article provides useful information on the microphysical properties of arctic clouds. My comments appear below.

General Comments

Line 159. "filtering".  Change to "identification and removal of shattered particle artifacts"

191-192. The Brown and Francis (1995) m(D) relationship has been shown to underestimate ice water content. (https://doi.org/10.1175/2010JAS3507.1, 10.1175/JAMC-D-22-0057.1). Could you possibly use a second m(D) relationship as well that would be more accurate?

232: How is $D_{eff}$ calculated? Does it include both liquid drops and ice particles?

In Figures 4 and 5, it might be good to put on the right side of each panel the approximate mean temperature with altitude.

Figure 4. I'd separate liquid and ice water contents.

Table 3. Separate ice and liquid water contents.

It might be helpful to modelers to have the PSD parameterized, as a gamma function. Also, show plots of the maximum measured particle diameter for each regime. Is the maximum diameter of the largest probe able to get the actual largest particles? Figure 7 with the PSD suggest that there are larger particles present but not measured.

246: is the air polluted or do you mean that there are fewer aerosols?

265: a stronger temperature inversion

Figure 6. This figure would be more interesting if you had two panels with separate panels for CDP and CIP+PIP data.

366. Right after Arctic. I strongly suggest having a figure with a schematic (pictorial) of the primary findings that would be simple to grasp.

**Minor Points**

Table 2 Year should be 2020, shouldn't it.

231: "to" to "with"

---

## Author Response (AR1)

Review of "Microphysical and thermodynamic phase analyses of Arctic low-level clouds measured above the sea ice and the open ocean in spring and summer", by Manuel Moser, Christiane Voigt, Tina Jurkat-Witschas, Valerian Hahn, Guillaume Mioche, Olivier Jourdan, Régis Dupuy, Christophe Gourbeyre, Alfons Schwarzenboeck, Johannes Lucke, Yvonne Boose, Mario Mech, Stephan Borrmann, André Ehrlich, Andreas Herber, Christof Lüpkes, and Manfred Wendisch, acp-2023-44.

**Response to reviewer 1**

Dear reviewer,
We are very grateful for your valuable feedback and suggestions which helped us to improve the manuscript. The manuscript has been thoroughly revised and point-by-point responses have been prepared. Please find below our replies, highlighted in blue, along with your suggestions. The revised manuscript is also provided with tracked-changes for clarity.

General comments:

Line 159. "filtering". Change to "identification and removal of shattered particle artifacts"

Has been changed. Additionally, we have added the new reference for SODA2.

Changes made in the manuscript:
- *(158) The data processing includes identification and removal of shattered particle artifacts, stuck bit correction, and particle sizing which is done with the processing software SODA (Software for OAP Data Analysis; Bansemer, 2023).*

191-192. The Brown and Francis (1995) m(D) relationship has been shown to underestimate ice water content. (https://doi.org/10.1175/2010JAS3507.1, 10.1175/JAMC-D-22-0057.1). Could you possibly use a second m(D) relationship as well that would be more accurate?

Thank you for the advice, we agree, the BF method is no longer up to date. Therefore, for the calculation of the CWC, we now use your suggested values (a = 0.00528 and b = 2.1). A direct comparison of the calculated CWCs shows that the new CWC is slightly higher than the old one:

[Figure]

Figure 1. Correlation of IWC calculated with mass dimension parameters given by Brown and Francis (1995) vs. new suggested IWC calculation.

We have adjusted all CWC-values in the manuscript and clarified which parameters for the mass-dimension relationship are used. Note that the definition of a cloud segment in Section 3.1 changes as it is defined by a CWC threshold. This changes for example the values in Table 3 minimally.

Changes made in the manuscript:
- Adjusted all CWC related values
- *(193) Formula (1) with D the particle diameter from the circle-fit method and the parameters (a = 0.00528 g cm-b and b = 2.1) proposed by 195 Heymsfield et al. (2010, 2023).*

In Figures 4 and 5, it might be good to put on the right side of each panel the approximate mean temperature with altitude.

We have added the median temperature including the 25th and 75th percentile in Fig. 4 and 5.

Changes made in the manuscript:
- Fig. 4 and 5, as well Fig. A1, A2, A3

Figure 4. I'd separate liquid and ice water contents.

Figure 4 is used to give a quick overview of the measured data, but for absolute values Table 3 and Fig. 5 are more suitable. We appreciate your suggestion to split the microphysics between ice and liquid water, so we have added additional rows in Table 3 in the manuscript that presents the values for liquid particles (assuming particles < 50 µm) and ice particles (assuming particles > 50µm). Corresponding plots following Figure 5 are given here:

[Figure]

Figure 2. Similar Figure as presented in Fig. 5 in the manuscript, but calculated for particles <50µm only (assumed as liquid droplets).

[Figure]

Figure 3. Similar Figure as presented in Fig. 5 in the manuscript, but calculated for particles >50μm only (assumed as ice particles).

In Chapter 3.2 we show how to separate cloud data into liquid, mixed phase and ice regimes. For the respective regimes, we created plots according to Fig. 5 including Tables with their respective microphysical properties and attached them to Appendix in the manuscript (requested by reviewer 2).

Changes made in the manuscript:
- (252) Table 3
- Added Tables A1, A2, A3, A4 and Figures A1, A2, A3

Table 3. Separate ice and liquid water contents.

Please see the answer to previous comment.

Changes made in the manuscript:
- (252) Table 3
- Added Tables A1, A2, A3, A4 and Figures A1, A2, A3

It might be helpful to modelers to have the PSD parameterized, as a gamma function. Also, show plots of the maximum measured particle diameter for each regime. Is the maximum diameter of the largest probe able to get the actual largest particles? Figure 7 with the PSD suggest that there are larger particles present but not measured.

We much appreciate the idea of adding gamma fits to the particle size distribution to enable analyses with other methods in the future. However fitting gamma distributions over the whole size range is challenging due to the multimodal shape. Therefore, the parameters for the gamma functions are given for the respective size range of each instrument. We have added the gamma functions in Fig. 7 including a Table in Appendix B1 showing the values of the fitted parameters. We are aware that there may be particles larger than the upper size limit of the PIP in the cloud regimes 1b, 2a, 2b,2c. However, we cover a particle size distribution with values distributed over more than 13 orders of magnitude. The influence of large particles, which exceed the size range of detection, is negligible in the calculated microphysical properties due to their very low number concentration.
We have added an explanation and the equation of the gamma fit in the manuscript.

Changes made in the manuscript:
- Added a gamma fit to Figure 7
- Added Table B2 showing the fitted gamma values
- *(321) In addition to the particle size distributions in Fig. 7, gamma functions are fitted over the sensitive size range of the respective instrument. Cloud particle size distribution usually follow gamma type functions of the form: (Formula 2) The fitted values for the dispersion μ, the slope λ and the intercept N0 are given in Table B1.*

246: is the air polluted or do you mean that there are fewer aerosols?

During cold air outbreaks in the Fram Strait airmasses from the central Arctic get transported to lower latitudes. These airmasses are typically exposed to fewer aerosol particles.
We have changed the sentence to: "In spring, cold air outbreaks with strong winds from the central Arctic bring dry air with a low aerosol load."

Changes made in the manuscript:
- *(263) In spring, cold air outbreaks with strong winds from the central Arctic bring dry air with a low aerosol load.*

265: a stronger temperature inversion

Adapted.

Changes made in the manuscript:
- *(286) In summer, warm and moist air advection from the south leads to a stronger temperature inversion and favors multilayer clouds (Eirund et al., 2020).*

Figure 6. This figure would be more interesting if you had two panels with separate panels for CDP and CIP+PIP data.

We have added a Figure in the Appendix similar to Fig. 6 but showing the $D_{eff}$ and N calculated for CDP/CAS, CIP and PIP respectively. Additionally, we show how the previous defined regimes are measured by the individual probes and calculate the proportion of the detected values compared to the combined values in Fig. 6.

Changes made in the manuscript:
- Added Figure B1

366. Right after Arctic. I strongly suggest having a figure with a schematic (pictorial) of the primary findings that would be simple to grasp.

We implemented this idea and added a schematic Figure summarizing the primary findings of this work in Section 4.

Changes made in the manuscript:
- Added Figure 10

232: How is $D_{eff}$ calculated? Does it include both liquid drops and ice particles?

Here the $D_{eff}$ calculation includes both, droplets and ice particles. Now that we distinguish between ice and water in the manuscript and in Table 3, the calculation of $D_{eff}$ is more clear.

Changes made in the manuscript:
- None, now clear with previous changes

Minor Points

Table 2 Year should be 2020, shouldn't it.

This typo was corrected in the manuscript.

Changes made in the manuscript:
- Table 2: *2020*

231: "to" to "with

Adapted.

Changes made in the manuscript:
- *(249) The largest differences of cloud properties are associated with the different seasons*

**Response to reviewer 2**

Dear reviewer,
We are very grateful for your valuable feedback and suggestions which helped us to improve the manuscript. The manuscript has been thoroughly revised and point-by-point responses have been prepared. Please find below our replies, highlighted in blue, along with your suggestions. The revised manuscript is also provided with tracked-changes for clarity.

Comments:

1. The main comment from the reviewer is about how Figures 4 and 5 and Table 3 are not separated into different cloud phases or different types of cloud hydrometeors. The reviewer recommends adding figure sub-panels similar to the original Figure 4 and 5, but separately plot the distributions of the 3 types of cloud thermodynamic phases (ice, mixed and liquid) as identified by the group number 1, 2 and 3 in Figure 6, respectively. This way the readers can see if the vertical distributions of these three cloud phases change between summer and spring, and between over ocean and sea ice. In addition, for table 3, more rows can be added to compare N, Deff, CWC, d_cloud_mean that are calculated for each of the 3 cloud phases separately.

Thank you for your comment. Figure 4 is intended to provide an overview of the collected cloud data and its distribution in altitude. For absolute values of the microphysical variables, Fig. 5 and Table 3 are much more suitable. We have created additional Figures and Tables similar to Fig. 5 and Table 3 with data from the ice, liquid, and water regimes, and have included them in the manuscript's appendix. Additionally, we have added new rows to Table 3, which give the cloud parameters calculated for small particles (<50μm, assumed to be water) and large particles (>50μm, assumed to be ice).

Changes made in the manuscript:
- (252) Table 3: added rows; Caption: *Table 3. Properties of Arctic low-level clouds (< 500 m) during AFLUX and MOSAiC-ACA for surface condition sea ice or ocean: Median number concentration N, median effective diameter ˜ D˜eff, median cloud water content CWC and mean horizontal cloud extent ˜ dcloud (calculated using the duration in cloud and mean aircraft speed, V = 60 m s-1). The values in the square brackets give the 25th and 75th percentile respectively. The microphysical properties are calculated from all detected cloud particles as well as for particles smaller than 50 μm (assumed to be liquid) and for particles larger than 50 μm (assumed to be ice). An asterisk indicates that a combination of two values within this column is not significantly different. These combinations are as follows: N: im-om, ˜ dcloud: ia-oa, ia-im, i-o, N˜<50μm: im-om, D˜eff,<50μm: ia-im, N˜>50μm: i-o.*
- Added Tables A1, A2, A3, A4 and Figures A1, A2, A3
- (237) *In addition to the microphysical cloud properties based on particles in the size range from 2.8 μm to 6.4 mm, the microphysical cloud properties for liquid particles (based on particles < 50 μm) and ice particles (based on particles > 50 μm) only are presented.*

2. This is a related question to comment 1, can the authors separate each instrument measurement into liquid or ice hydrometeor? In Figure 6, the Deff and N values seem to use the combined measurements of multiple probes. It would be helpful to trace back how these ice, mixed and liquid phase 1-Hz measurements are contributed by individual probes. For example, the authors can add sub-panels to this Figure 6, using CAS, CDP, CIP and PIP, individually, and then calculate just the Deff and N for that probe alone (in a limited size range), and show how their own Deff vs N would look, color code where the 1-Hz samples of ice, mixed and liquid exist in that probe's measurement space. This way one can understand how individual probes contribute to the groups 1, 2 and 3 of cloud phases in the combined Deff vs N plot.

Another suggestion is, if the authors can separate the ice and liquid hydrometeors within one second, then the authors can calculate the N, Deff, CWC just for liquid or just for ice hydrometeors. Note that this type of calculation wouldn't have the mixed phase because it is based on the type of hydrometeors, not the type of cloud segment. The reviewer would like to point to some previous methods of defining ice and liquid from CDP, 2DC and 2DS probes in the following papers. These probes have some similar size range to the ones used in this study.

Yang, C.A., M. Diao, A. Gettelman, K. Zhang, J. Sun, W. Wu, G. McFarquhar, Ice and Supercooled Liquid Water Distributions over the Southern Ocean based on In Situ Observations and Climate Model Simulations, Journal of Geophysical Research: Atmosphere, 126, e2021JD036045. https://doi.org/10.1029/2021JD036045, 2021.

D'Alessandro, J., M. Diao, C. Wu, X. Liu, B. Stephens, and J.B. Jensen, "Cloud phase and relative humidity distributions over the Southern Ocean in austral summer based on in-situ observations and CAM5 simulations", J. Climate, doi:10.1175/JCLI-D-18-0232.1, 2019.

Regarding the first part of comment 2: To demonstrate how each individual cloud probe contributes to the regimes in the $D_{eff}$ - N space, we have included a Figure in the appendix, similar to Fig. 6, but now display the $D_{eff}$ and N values calculated for CDP/CAS, CIP, and PIP, respectively. Moreover, we show how the previously defined regimes are measured by the individual probes and determine the proportion of the detected values compared to the combined values in Fig. 6.

Regarding the second part of comment 2: The two references cited demonstrate a very effective method for differentiating between ice and liquid water within a second. Although we appreciate these works and methods, within one second we now differentiate between ice and water based on the particle sizes, as doing otherwise would be beyond the scope of our study. We have added additional columns to Table 3, as mentioned in our response to comment 1. However, we will address the proposed methods in future publications. We are also happy to contribute on an individual basis, if an analysis using the mentioned method is requested.
The given papers are now cited in the manuscript for referring to a different method in order to distinguishing between liquid and ice.

Changes made in the manuscript:
- Added Figure B1
- Added columns in Table 3
- *(195) Another effective method to separate the liquid and ice fraction in clouds is recommended by D'Alessandro et al. (2019). The method classifies the thermodynamic phase of the cloud into ice, liquid or a mixed-phase based on a combination of microphysical properties recorded by similar in-situ cloud particle sizing instruments (Yang et al., 2021). In*

*this work however the thermodynamic phase discrimination in Section 3.2 is achieved with the PN.*

3. Because the Deff and N currently used in Section 3.1 seems to be a combined value of both ice and liquid, the reviewer suggests the description of higher or lower Deff not be directly referring to liquid droplets. Line 280, the author said "also during summer a decrease of Deff of the liquid droplets is observed". This is more likely because there are more liquid droplets in summer which tend to be smaller than ice, not because these liquid droplets in summer are smaller than the liquid droplets in spring.

In the new manuscript we are not directly referring to liquid droplets anymore but changed the sentence to: "Also during summer when clouds are most likely in a liquid state a decrease of $D_{eff}$ with altitude is observed."
As we have additionally separated the microphysical properties in Table 3 into liquid and ice, the interpretation should now be facilitated.

Changes made in the manuscript:
- *(303) Also during summer when clouds are most likely in a liquid state a decrease of Deff with altitude is observed.*

4. Comparisons between summer and spring and between ocean and sea ice need some more statistical significance tests. For instance, the analysis in Figure 4, the whiskers represent 97.5 and 2.5th percentile, and Table 3 has 75 and 25th But it would be helpful to provide a t-test to check if the two groups of data (summer vs spring, or ocean vs sea ice) are significantly different statistically.

Thank you very much for your suggestion to test the differences in the data using significance tests. We have checked interesting combinations for the significance of the differences in each row in Table 3: We used the t-test only for the mean cloud extend. For the other microphysical cloud data, we used the Wilcoxon test since we compare median values and the data are not normally distributed.
With the conducted tests, we found that the horizontal mean cloud extend does not change significantly for all combinations in different environmental conditions.
However, the tests reveal the total difference for both seasons is still valid.
We also discovered a non-significant difference in the variable N through the tests. Specifically, we found that in the summer season, the difference in N over ice and ocean is not significant.
The tests were also performed on the variables for cloud data calculated from liquid and ice particles only, which are added to Table 3, as well for similar Tables added to the Appendix. However here the increased number concentration of particles in the liquid regime over the sea ice compared to the ocean in summer is statistically significant. We have incorporated these new findings into the manuscript.

Changes made in the manuscript:
- (252) Table 3 caption: *Table 3. Properties of Arctic low-level clouds (< 500 m) during AFLUX and MOSAiC-ACA for surface condition sea ice or ocean: Median number concentration N, median effective diameter ˜ D˜eff, median cloud water content CWC and mean horizontal cloud extent dcloud (calculated using the duration in cloud and mean aircraft speed, V = 60 m s-1). The values in the square brackets give the 25th and 75th percentile respectively. The microphysical properties are calculated from all detected cloud particles as well as for particles smaller than*

*50 μm (assumed to be liquid) and for particles larger than 50 μm (assumed to be ice). An asterisk indicates that a combination of two values within this column is not significantly different. These combinations are as follows: N: im-om, dcloud: ia-oa, ia-im, i-o, N<50μm: im-om, Deff,<50μm: ia-im, N>50μm: i-o.*

- *(239) In order to determine whether two values within a single column in Table 3 are statistically different, we conducted T-tests for the mean values and Wilcoxon tests for the medians. The significance level was set at 5% to decide whether the prevalent environmental conditions influence the properties of the clouds. We examined the following combinations for each property value within a row: Between the surface condition sea ice (i) and ocean (o) in spring (a) and in summer (m) (ia-oa, im-om), between spring and summer for the two surface conditions (ia-im, oa-om), as well as between the cloud data for each season (a-m) and surface condition (i-o). In case there is a combination for which the difference is not statistically significant, it is marked with an asterisk in Table 3, and the corresponding combination is indicated in the caption. For example, the asterisk in the first row indicates that there is no significant difference in the data between the N we observe for clouds over sea ice compared to cloud over the ocean during the summer campaign.*
- *As well statistical tests for Table in the Appendix*
- *(275)* The influence of different surface conditions on the horizontal cloud extension does not appear to be significant in our data.
- *(295) This process could also explain the reduction of N over the ocean, which is significantly observed in spring. However, in Sect. 3.2 we will show, that the differences of N measured over the sea ice and the open ocean might result from different aerosol sources.*
- *(410) The increased N observed in mixed-phase clouds, in aerosol particles, in liquid clouds and in the total cloud particle measurements above the sea ice may be explained by surface processes emitting sea salt.*

5. Minor typos. Line 376, We observe lager (should be larger) ice…

Adapted.

Changes made in the manuscript:
- *(407) We observe larger ice crystals in spring and smaller liquid droplets in summer conditions.*

6. Line 394, typo Figures have been design (should be designed).

Adapted.

Changes made in the manuscript:
- *(431) Figures have been designed with the python software Pylustrator*

7. In data availability, there is currently no description about where the AMSR2 satellite data for sea ice coverage during the AFLUX and MOSAIC-ACA campaigns are stored. It would be helpful for follow-up studies if the authors can provide other data used in this analysis, such as satellite and HYSPLIT back trajectory data.

The AMSR2 derived sea ice coverage is available from the University of Bremen (https://seaice.uni-bremen.de/). However, extracted sea ice coverage along the flight path of Polar 5 is available via the python package ac3airborne.
We extended line 392 in the data availability section by the sentence. "The data ac3airborne package provides as well access to sea ice coverage along the flight path extracted from data available at University of Bremen ( https://seaice.uni-bremen.de/)."

The HYSPLIT model is an online tool that does not require any specific input data by the user. The user just needs to specify time, position, atmospheric model and starting altitude. It is freely available at https://www.ready.noaa.gov/HYSPLIT.php.
We added to the data availability section "The HYSPLIT model is a freely accessible online tool available at https://www.ready.noaa.gov/HYSPLIT.php."

Changes made in the manuscript:
- *(422) Processed in-situ data from the AFLUX and MOSAiC-ACA campains are freely available via the world data center PANGAEA (Moser and Voigt, 2022; Moser et al., 2022; Dupuy et al., 2022a, b). Data can be easily reproduced and analyzed by the python package ac3airborne (Mech et al., 2022b) including a package for flight segmentation (Risse et al., 2022), where each research flight is split up into logical parts like ascends, descends, specific patterns for in-situ probing, etc.. The data ac3airborne package provides as well access to sea ice coverage along the flight path extracted from data available at University of Bremen (https://seaice.uni-bremen.de/). Raw in-situ cloud data recorded by the CAS, CAS, CIP and PIP are archived at the German Aerospace Center and are available on request. Raw data by the PN and 2D-S are available from R. Dupuy (regis.dupuy@uca.fr) on request. The HYSPLIT model is a freely accessible online tool available at https://www.ready.noaa.gov/HYSPLIT.php. Figures have been designed with the python software Pylustrator (Gerum, 2020).*